# Supracellular organization confers directionality and mechanical potency to migrating pairs of cardiopharyngeal progenitor cells

Yelena Y Bernadskaya[1]*[†], Haicen Yue[2†], Calina Copos[3], Lionel Christiaen[1,4,5]*[†], Alex Mogilner[2]*[†]

[1]Center for Developmental Genetics, Department of Biology, New York University, New York, United States; [2]Courant Institute of Mathematical Sciences and Department of Biology, New York University, New York, United States; [3]Mathematics and Computational Medicine, University of North Carolina at Chapel Hill, Chapel Hill, United States; [4]Sars International Centre for Marine Molecular Biology, Bergen, Norway; [5]Department of Heart Disease, Haukeland University Hospital, Bergen, Norway

*For correspondence:
yb372@nyu.edu (YYB);
lc121@nyu.edu (LC);
mogilner@cims.nyu.edu (AM)

[†]These authors contributed equally to this work

Competing interest: The authors declare that no competing interests exist.

**Abstract** Physiological and pathological morphogenetic events involve a wide array of collective movements, suggesting that multicellular arrangements confer biochemical and biomechanical properties contributing to tissue-scale organization. The *Ciona* cardiopharyngeal progenitors provide the simplest model of collective cell migration, with cohesive bilateral cell pairs polarized along the leader-trailer migration path while moving between the ventral epidermis and trunk endoderm. We use the Cellular Potts Model to computationally probe the distributions of forces consistent with shapes and collective polarity of migrating cell pairs. Combining computational modeling, confocal microscopy, and molecular perturbations, we identify cardiopharyngeal progenitors as the simplest cell collective maintaining supracellular polarity with differential distributions of protrusive forces, cell-matrix adhesion, and myosin-based retraction forces along the leader-trailer axis. 4D simulations and experimental observations suggest that cell-cell communication helps establish a hierarchy to align collective polarity with the direction of migration, as observed with three or more cells in silico and in vivo. Our approach reveals emerging properties of the migrating collective: cell pairs are more persistent, migrating longer distances, and presumably with higher accuracy. Simulations suggest that cell pairs can overcome mechanical resistance of the trunk endoderm more effectively when they are polarized collectively. We propose that polarized supracellular organization of cardiopharyngeal progenitors confers emergent physical properties that determine mechanical interactions with their environment during morphogenesis.

## Introduction

Cell migration is a fundamental cellular behavior involved in developmental and physiological processes including germline, craniofacial, and cardiac development, angiogenesis and wound healing, and pathogenesis such as cancer metastasis (*Rørth, 2009*; *Scarpa and Mayor, 2016*). In complex dynamic multicellular environments, cells integrate biochemical and mechanical cues to guide their migration. Some migration specialists, like neutrophils, navigate complex environments as single cells (*Wang et al., 2020*). Conversely, many developmental, homeostatic, and pathogenic morphogenetic events involve the coordinated movements of cellular collectives, as observed during

neural crest migration in chick, lateral line migration in zebrafish, and border cell migration in the *Drosophila* ovary (*Piacentino et al., 2020*; *Olson and Nechiporuk, 2018*; *Peercy and Starz-Gaiano, 2020*). The properties that emerge from collective organization are thought to facilitate biochemical and mechanical integration and foster efficient and accurate tissue morphogenesis in a multicellular context (*Malet-Engra et al., 2015*; *Theveneau et al., 2010*; *Shellard et al., 2018*; *van Helvert et al., 2018*; *Friedl and Mayor, 2017*).

Migratory collectives typically exist on a continuum with varying degrees of cell-cell contact and polarity (*Capuana et al., 2020*; *Mayor and Etienne-Manneville, 2016*). In minimally differentiated groups, individual cells move as autonomous units, while adjusting directionality and speed relative to their neighbors (*Szabó et al., 2006*). At the other extreme of collective organization, cells are integrated into supracellular arrangements, with marked front-to-back specialization and continuity of cytoskeletal structures between neighboring cells, ensuring mechanical coupling (reviewed in *Shellard and Mayor, 2019*). Such collective polarity implies communication between cells to coordinate subcellular processes.

Numerous studies uncovered mechanisms underlying collective organization, such as contact inhibition of locomotion (CIL) (*Mayor and Etienne-Manneville, 2016*; *Ebnet et al., 2018*) and leader-mediated inhibition of protrusive activity in follower cells (*Cai et al., 2014*; *Serwane et al., 2017*). Ultimately, both biochemical and mechanical properties of migratory collectives contribute to successful tissue morphogenesis. While many biochemical aspects of cell migration have been investigated, measurement of in vivo mechanical forces involved in morphogenetic cell migration has been a challenge (*Campàs et al., 2014*). To understand the mechanics of collective locomotion, in vitro techniques such as traction force microscopy were developed and used to correlate forces with movement and cytoskeletal dynamics (*Danuser and Waterman-Storer, 2003*). In more complex embryo settings, the distribution of mechanical forces can be inferred from imaging datasets, combined with available direct measurements of membrane tension (*Campàs et al., 2014*; *Veldhuis et al., 2017*; *Godard et al., 2020*). As a complement to biophysical measurements, computational modeling offers a powerful option to reverse-engineer forces from observed cell shape and enable in silico predictions that can be compared to experimental observations (*Godard et al., 2020*; *Sherrard et al., 2010*). There is a rich inventory of modeling approaches, from simple conceptual models of cells as point-like persistent walkers interacting with distance-dependent forces (*Méhes and Vicsek, 2014*), to detailed continuous or discrete models of interacting cells as distributed mechanical objects with complex rheology and free boundaries (*Winkler et al., 2019*; *Buttenschön and Edelstein-Keshet, 2020*; *Alert and Trepat, 2020*).

We use the cardiogenic lineage of the tunicate *Ciona*, a simple chordate among the closest relatives of vertebrates (*Dehal and Boore, 2006*; *Putnam et al., 2008*), to develop and test a computational model of collective polarity and directed cell migration. During *Ciona* embryonic development, the cardiopharyngeal precursors are born as two superficially equivalent cells arising from the division of bilateral founders. The resulting cells migrate from their origin in the tail to the ventral trunk, hence their denomination as trunk ventral cells (aka TVCs) (*Christiaen et al., 2008*; *Gline et al., 2015*; *Bernadskaya et al., 2019*). The TVCs migrate as pairs, offering the simplest possible model of polarized collective cell migration. The anterior leader cell extends dynamic lamellipodia-like protrusions, while the posterior trailer terminates in a tapered retractive edge (*Christiaen et al., 2008*). Under unperturbed conditions, the migrating TVCs remain committed to their leader/trailer positions (*Gline et al., 2015*; *Bernadskaya et al., 2019*). TVCs contact multiple tissues during migration, including the posterior mesenchyme, the trunk endoderm, and the ventral epidermis, which serves as substrate (*Gline et al., 2015*). TVCs maintain polarized organization along the anterior posterior axis and spheroid shapes as they invade the extracellular space between the epidermis and the endoderm. *Ciona* TVCs thus represent a simple and intriguing model to study the mechanics and polarity of cells migrating in an embryonic context (*Figure 1A*).

Here, we model TVC shape and behavior using the Cellular Potts Model (CPM) (*Thüroff et al., 2019*; *Rens and Edelstein-Keshet, 2019*; *Fortuna et al., 2020*). In the CPM framework, each cell is a shifting shape described by a sum of mechanical energies of cell-substratum and cell-cell adhesions, surface tension and hydrostatic pressure, and protrusion and retraction forces (*Figure 1A*, bottom panel). These energies effectively correspond to realistic cytoskeleton-generated forces (*Rens and Edelstein-Keshet, 2019*; *Sherrard et al., 2010*). The cell boundaries fluctuate, mimicking random

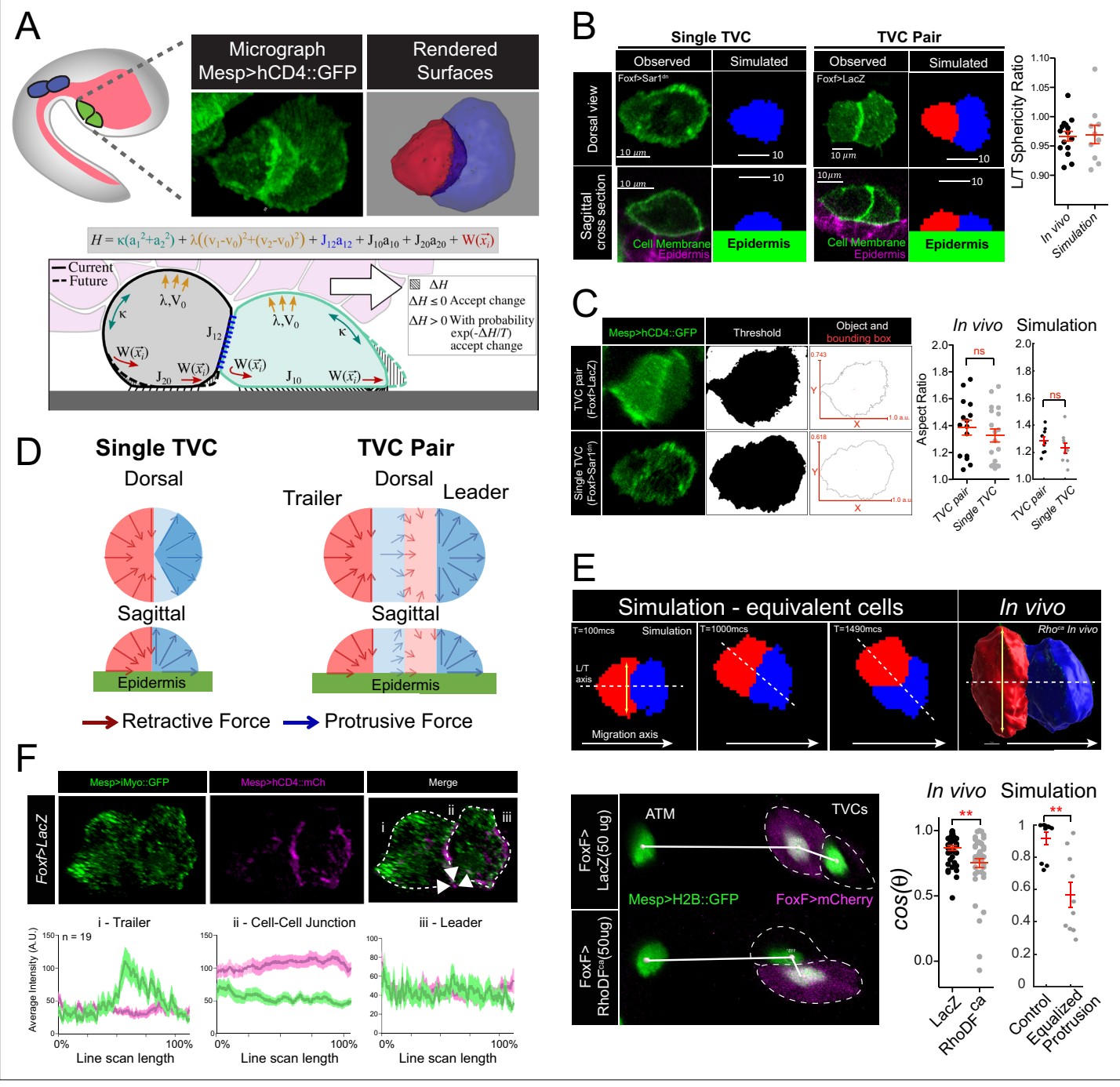

**Figure 1.** Model of force distribution in migrating trunk ventral cell (TVC) pairs. (**A**) Diagram of *Ciona robusta* embryo at the late tailbud stage (embryonic stage 23). Migrating TVCs are shown in green, their non-migratory sister cells, anterior tail muscles (ATMs), in blue. The endoderm is shown in pink. A micrograph of a migratory pair of TVCs is shown with the leader to the right and the trailer to the left. Cell membranes are marked with *Mesp>hCD4::GFP*. To the right is a surface-rendered image of the migratory TVC pair with leader in blue and trailer in red. Schematic diagram showing the mechanical parameters related to cells' movement and morphology, reflecting volume conservation (yellow), surface tension (green), cell-cell adhesion (blue), cell-epidermis adhesion (black), and active protrusion/retraction forces (red). The cell pair moves to the right, with the green cell as the leader cell and the gray cell as the trailer cell. Overlying endoderm cells are shown in pink; the underlying epidermis in gray. The shape change (shaded area) is accepted or rejected depending on the energy change $\Delta H$ related to it. The equation above shows the effective mechanical energy, $H$, of the cell pair. The meaning of the parameters is explained in the text. (**B**) Comparison of cell shape in the experiment and in simulation for single migrating cell and migrating cell pair. Scatter plot shows ratio of leader to trailer sphericity derived from in vivo measurements and in simulations. In vivo data were pooled from two biological replicates. No statistical difference was identified by Student's *t*-test between the in vivo and simulated data. Micrographs

*Figure 1 continued on next page*

*Figure 1 continued*

show dorsal and lateral view of 3D images of TVC. TVC membranes are marked with *Mesp>hCD4::GFP,* and epidermal cell membranes are marked with *Mesp>hCD4::mCherry*. (**C**) Aspect ratios of migrating cell pairs compared to aspect ratios of single migrating cells calculated in Fiji and in simulations. Red lines show length of bounding box width and heights normalized to the width. Scatter plots show mean with standard error of in vivo and simulated data. Statistical analysis was performed using Student's *t*-test. No significant difference between conditions in vivo and in simulation. Data are pooled from two biological replicates. (**D**) Dorsal and sagittal views of force distributions within a single cell (left) and two connected cells (right) in our model for unperturbed cells. Arrow thickness indicates relative strength and direction of force. Cell anterior is in blue and posterior in red. (**E**) Simulation and in vivo verification of equalized protrusion in leader and trailer. Top panels show results of simulated cell positions at indicated time points and the morphology of an in vivo cell pair when trailer protrusion is upregulated by expression of constitutively active Ras (*Ras^ca^*). Solid arrows show the direction of migration. Bottom panels show representative positions of migrating cells with respect to the stationary ATMs. Graphs show the cosine of the radian angle of the leader/trailer axis to the axis of migration derived from in vivo and simulations. Inheritance of the perturbing plasmid is followed using the cytoplasmic marker *FoxF>mCherry* (magenta), and the nuclei of the TVCs and ATMs is marked with *Mesp>H2B::GFP* histone marker. In vivo data were pooled from two biological replicates. Statistical analysis was performed using Student's *t*-test for the experimental data and Student's *t*-test with Welch's correction for the simulation data, ** $p<0.01$. In simulations here and below, time is measured in units of Monte Carlo step (mcs). (**F**) Distribution of myosin reporter iMyo-GFP intensity compared to membrane marker *Mesp>hCD4::mCherry*. Dashed arrows on the merged micrograph indicate the directionality of the line scan, which moves in the direction of the arrow. Mean values with standard error are plotted on the graphs. Data were pooled from two biological replicates.

The online version of this article includes the following figure supplement(s) for figure 1:

**Figure supplement 1.** In silico analysis of evolution of single-cell shape and two-cell cohesion under differing force distribution.

**Figure supplement 2.** Sphericity of leader and trailer cells.

**Figure supplement 3.** Myosin distribution at the cell-cell junction.

force and movement on subcellular scale, and shape changes minimizing the total energy are accepted. This results in evolving, collectively moving cells (*Videos 1 and 2*). CPM is advantageous in modeling the TVC pair as it allows us to reproduce both the detailed evolving 3D shapes of motile cells and the deformation of tissues surrounding these cells over a reasonable computational time (*Thüroff et al., 2019*; *Rens and Edelstein-Keshet, 2019*; *Fortuna et al., 2020*) – one of the more challenging tasks for detailed force-balance models in 3D (*Wu et al., 2018*).

By examining the distribution of forces required to recapitulate the shape of migrating TVCs in silico, we first predict the polarized distributions of protrusive activity, cell-matrix adhesion, and acto-myosin contractility across cells, and test these predictions using in vivo observations and molecular perturbations. We propose that the leader and trailer cells form the simplest possible supracellular arrangement of a migratory collective. We hypothesize that this arrangement emerges from a leader-trailer mode of migration, which invokes polarized abilities to respond to extracellular guidance and mutual cell-cell attraction. Our model explains the preference for a linear arrangement of cells polarized in the direction of migration as this arrangement improves the persistence of migrating cells, which can presumably better buffer variations in migration cues. Finally, our model predicts that the linear arrangement of cardiopharyngeal progenitors allows them to distribute forces in a way that helps them deform the trunk endoderm and facilitates their migration despite the mechanical resistance exerted by the developing gut primordium.

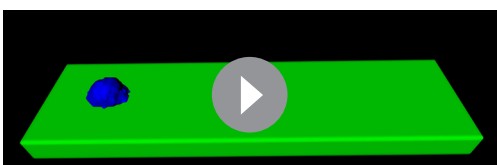

**Video 1.** In silico model of a single migrating trunk ventral cell (TVC).
https://elifesciences.org/articles/70977/figures#video1

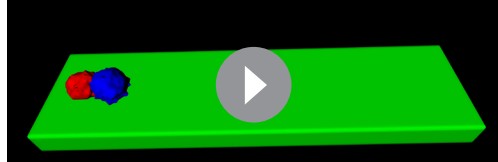

**Video 2.** In silico model of trunk ventral cell (TVC) pair (leader in blue, trailer in red) migrating along a surface.
https://elifesciences.org/articles/70977/figures#video2

## Results

## Polarized protrusive and retraction forces are distributed across a supracellular collective

Cell morphology reflects and conditions cellular behavior, inasmuch as both emerge from underlying mechanical forces (*Figure 1A*; *Mogilner and Keren, 2009*). From that standpoint, cell shape provides a phenomenological proxy to biophysical forces driving cellular behavior (*Campàs et al., 2014*; *Maitre, 2017*). In migrating collectives, leader cells typically adopt splayed morphologies with protrusive activity at the leading edge, while trailing cells display a tapered rear (*Nabeshima et al., 1995*). This organization is conspicuous in pairs of multipotent cardiopharyngeal progenitor cells (aka TVCs; *Figure 1A*) in the embryo of the tunicate *Ciona*. TVC pairs migrate along a stereotypical path, canalized by surrounding tissues, while maintaining cell-cell junctions and polarizing along the direction of migration: the leader cell generates a protrusive edge, while the retracting trailer cell has a tapered rear and higher sphericity (*Christiaen et al., 2008*; *Gline et al., 2015*; *Bernadskaya et al., 2019*; *Figures 1A and 2A*, *Figure 1—figure supplement 2*). The *Ciona* TVCs thus provide the simplest possible example of directional migration of a polarized cell collective.

We leveraged the TVCs' simplicity and experimental tractability, combined with mathematical modeling, to simulate cell shape and behavior from first biophysical principles. We used a previously established experimental perturbation, whereby mosaic expression of a dominant-negative inhibitor of the secretory pathway (Sar1$^{dn}$) stalls the migration of the transfected TVC while allowing the other cell to migrate on its own, and compare the shapes of single cells to those of cell pairs (*Gline et al., 2015*). Single migrating TVCs generally display morphologies intermediate to those of either leader or trailer, generating both a leading edge and a retracting rear end, thus producing overall shapes similar to migrating pairs (*Figure 1B and C*). We further quantified the similarity of cell shapes by comparing the aspect ratios, defined as the ratio between the length and width of the smallest rectangular bounding box that can enclose the cell or cell pair, of single migrating TVCs with TVC pairs. This comparison shows that both maintain similar overall shapes (*Figure 1C*), leading us to hypothesize that similar force distribution profiles may exist in both conditions.

CPM (*Thüroff et al., 2019*; *Rens and Edelstein-Keshet, 2019*; *Fortuna et al., 2020*) predicts that cell morphologies emerge from the mechanical energies of diverse force-generating processes distributed within each cell (*Figure 1A*, bottom panel). To explore the cell-autonomous mechanics responsible for the observed shapes of one- and two-cell systems that we can observe in vivo, we selected parameters characterizing the adhesion, cortex tension, and hydrostatic forces from general considerations that apply to most motile cells (see Materials and methods), and investigated the spatial distribution of protrusive and retractive forces that would best recapitulate observed cell shapes as indicated in *Figure 1B*. We first focused on the spatial-angular distribution of protrusive and retractive forces in single cells. By varying the width of the angular segment for retraction forces at the rear, we observed that narrowly focused retractive forces cause aberrant widening of the leading edge and fail to produce the tapered rear (*Figure 1—figure supplement 1A*, bottom row), suggesting that a broadly retracting trailing edge better accounts for the observed shapes of single cells.

In contrast, a wide protrusive force distribution widens and flattens the leading edge (*Figure 1—figure supplement 1A*), recapitulating the observed shapes (*Figure 1B*). In general, several combinations of protrusive and retractive force distributions produce cell shapes that qualitatively recapitulate observations (*Figure 1B*, *Figure 1—figure supplement 1A*). Simulations showed that the single-cell shape is faithfully reproduced if the retraction force is centripetal, radially converging to the cell center along the width and heights of the rear half of the cell. Similarly, protrusion force is centrifugal in the front half of the cell, diverging from the cell center (*Figure 1D*, *Video 1*). Other force distributions tested in the cell are discussed in Appendix 1. The polarized distributions of protrusive and retractile forces at the leading edge and trailing rear, respectively, are thus consistent with classic models of single-cell migration on two-dimensional substrates (*Ridley et al., 2003*).

Empowered by our single-cell simulations, we turned to modeling polarized cell pairs. Although single TVC migration (*Gline et al., 2015*) suggests that individual TVCs are migration-competent and do not require a cell partner, computationally stitching two identical single-cell models failed to reproduce the polarized morphology of migrating pairs. Specifically, when simulated leader and trailer cells were endowed with equivalent protrusive activity, the width of the trailer front extended beyond the width of the leader posterior, a morphology not observed in control embryos (*Figure 1E*, yellow

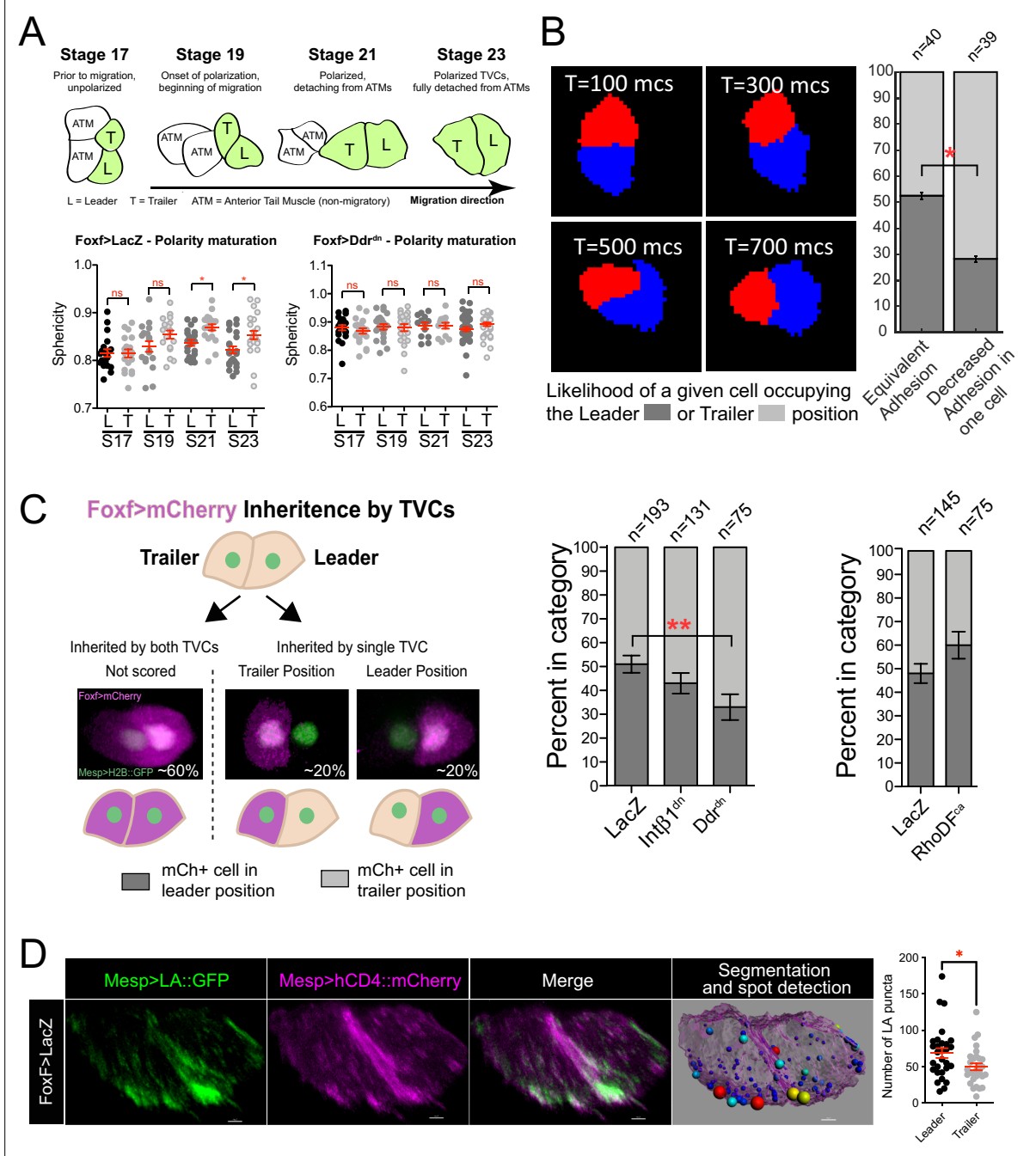

**Figure 2.** Polarized matrix adhesion promotes adoption of leader/trailer cell state. (**A**) Establishment of leader/trailer polarity as measured by the asymmetry that develops between leader and trailer sphericity as cells polarize in the direction of migration. Diagram depicts dorsal view of cells at stages when sphericity was calculated. Migratory cells are highlighted in green. L, leader; T, trailer; ATM, anterior tail muscle. Scatter plots show mean with standard error. Data were pooled from three biological replicates. Statistical significance tested using ANOVA followed by Bonferroni test to compare means. *p<0.05 (**B**) Simulation of decreasing extracellular matrix (ECM) adhesion in one cell (red) of a migrating cell pair. Cells are migrating to the right starting in a parallel orientation as shown in T = 100 Monte Carlo steps (mcs). Bar graphs show likelihood of either cell assuming the leader or trailer position in either control conditions (50/50 likelihood) or when adhesion in red cell is decreased. Standard error of proportion is shown, and statistical analysis of the proportions is done using Fisher's exact test. (**C**) In vivo modulation of ECM adhesion using mosaic inheritance of the *Foxf>Intβ1^dn^*, *Foxf>Ddr^dn^*, and *Foxf>RhoDF^ca^*, marked by *Foxf>mCherry*. Diagram shows a schematic of mosaic inheritance of transgenes and resulting distribution of mCherry fluorescence. Bar graphs show likelihood of cell that inherits the transgenic constrict to be found in either leader or trailer position. Data were pooled from three biological replicates. Error bars are standard error of proportion. Statistical analysis was performed using Fisher's exact test. (**D**) Micrographs show distribution of Mesp>Lifeact::GFP in leader and trailer cells. Image on the right shows a representative rendered cell

*Figure 2 continued on next page*

*Figure 2 continued*

pair surface with spot detection based on 10% highest GFP intensity. Spots are color-coded from smallest (blue) to largest (red). Scatter plot on the right shows number of GFP puncta per cell. Data were pooled from two biological replicates. Statistical analysis was performed using Student's *t*-test. *p<0.05.

The online version of this article includes the following figure supplement(s) for figure 2:

**Figure supplement 1.** Acute reduction of extracellular matrix (ECM) adhesion in single cell (red) causes detachment of that cell from the underlying epidermis and recapitulates the phenotype observed in vivo (bottom right) with the detached cell positioned on top of the cell that maintains ECM adhesion.

two-headed arrow). This also impacted collective polarity, causing the simulated trailer to leave its posterior position and travel side-by-side with the leader (*Figure 1E*). We quantified this phenomenon by calculating *cos(θ)* of the angle at which the leader/trailer axis intersects the direction of migration (*Figure 1E*). The predicted leader-dominated protrusive activity of the cell pair was consistent with previous observations that the typical leader TVC has a wide leading edge with lamellipodia-like protrusions that depend on Rhod/f- and Cdc42-controlled actin networks (*Christiaen et al., 2008*).

Remarkably, we could reproduce this predicted behavior in vivo by using the TVC-specific minimal *Foxf* enhancer to misexpress a constitutively active form of Rhod/f mosaically in either the prospective leader, trailer, or both cells, and measure the angle between the TVCs and their stationary sister cells, the anterior tail muscles (*Christiaen et al., 2008*; *Beh et al., 2007*) (ATMs) (*Figure 1E*). Under these conditions, experimentally increasing protrusive activity in one or both migrating cells disrupted their collective polarity, with cells more likely to migrate in parallel as predicted by in silico simulations (*Figure 1E*). This result suggests that the protrusive activity is suppressed in trailer cells compared to leader and single cells.

Nevertheless, simulations that reduce protrusive activity in the trailer without increasing its retractile forces cause the leader to detach and move forward on its own, overcoming significant mutual adhesion between the two cells (Figure 1—figure supplement 1C). This suggests that some protrusive activity in the trailer is still required, and that, in paired TVCs, the cells exert equivalent forces, while coordinating their activities to distribute protrusive and retractile forces to the leader and trailer cells, respectively (*Figure 1—figure supplement 1B–E*).

Next, we sought to further probe the supracellular model using phenomenological observations. Notably, the best computational recapitulation of observed shapes was achieved with centripetal retractive forces dominating in the trailer, pulling the cell rear forward and down, while the centrifugal protrusive forces in the leader pushed the front forward (*Figure 1B and D*, *Video 2*). The morphology produced by this force distribution also reproduced the measured aspect ratio of the motile TVC pair and single cells (*Figure 1C*), as well as asymmetries of the leader and trailer as reported by the ratio of their sphericities in vivo compared to simulations (*Figure 1B*). This is reminiscent of the centripetal character of actomyosin contractility (*Yam et al., 2007*), which led us to analyze myosin distribution in migrating cell pairs. Consistent with the model's prediction of a trailer-polarized retractile activity, we observe that iMyo-GFP, an intrabody that recognizes non-muscle myosin II through a conserved epitope (*Chaigne et al., 2013*; *Hashimoto et al., 2015*), accumulates at the rear of the trailer cell and is relatively depleted from the leader-trailer junction (*Figure 1F*, *Figure 1—figure supplement 3*), implying that the retraction in the trailer is dominant, while the retractive force in the leader is weak, lending further support to the hypothesis that TVC pairs migrate as a supracellular collective.

Taken together, our computational predictions and experimental observations support a model where pairs of multipotent cardiopharyngeal progenitors migrate as a polarized supracellular collective, with protrusive activity and myosin-based retraction distributed across leader and trailer cells, respectively.

## Polarized cell-matrix adhesion contributes to leader/trailer states of migrating cells

Cell-matrix interactions and distribution of protrusive activity to the leading TVC result in the generation of a broad leading edge during the establishment of collective TVC polarity (*Bernadskaya et al., 2019*). The flattened leading edge of the leader lowers the cell's overall sphericity. Conditions that perturb cell-matrix adhesion often increase the sphericity of leading cells, suggesting that they fail to establish flattened protrusions at the leading edge (*Bernadskaya et al., 2019*). We compared the

sphericity of the leader and trailer cells starting with TVC birth at developmental stage 19 prior to migration (*Figure 2A*). TVCs' sphericities did not differ at stage 19, suggesting that they are born with equivalent shapes. Cell sphericities begin to differ significantly during migration, indicating establishment of leader/trailer polarity, adoption of distinct states (*Figure 2A*), and orientation in direction of migration. Collective TVC polarization is abolished by disruption of cell-matrix adhesion by misexpression of a dominant-negative version of the collagen receptor, Discoidin domain receptor (Ddr$^{dn}$), which alters integrin-mediated cell-matrix adhesion and disrupts polarized Bmp-Smad signaling (*Bernadskaya et al., 2019*). This suggests that collective polarity and anisotropy of actin dynamics actively mature in response to extracellular cues and that leader/trailer selection can be biased based on the relative amount of cell-matrix adhesion experienced by the TVCs.

We previously showed that integrin ß1 (Intß1)- and Ddr-mediated signaling and cell-matrix adhesion to the basal lamina of the ventral trunk epidermis promote collective polarity and directional movement of TVC pairs (*Bernadskaya et al., 2019*; *Figure 2A*). Here, we harnessed the predictive power of our model to explore the consequences of varying the distribution and strength of cell-matrix adhesion forces across the cell pair in silico. We begin by simulating two cells side-by-side with the same protrusive/retractive force distribution, but with cell-matrix adhesion in one of the cells lower than the other. In order to focus on the cell-autonomous behaviors specific to the TVCs, we performed these simulations without modeling the overlying endoderm under which the cells move in vivo. In these conditions, the cell with reduced adhesion was more likely to assume the trailer position (*Figure 2B*). We tested this prediction in vivo using mosaic overexpression of the dominant negative forms of Intß1 and Ddr driven by the *Foxf* minimal TVC enhancer, tracked by co-inheritance of the mCherry marker (*Figure 2C*). In control mosaic embryos, *Foxf*$^{TVC}$-driven fluorescent protein expression marks either the leader or trailer cell in equal proportions, consistent with previously published data (*Gline et al., 2015*; *Figure 2C*). In contrast, coexpression of *Intß1*$^{dn}$ increased the proportion of labeled trailer cells to 57% in mosaic embryos (*Figure 2C*). Cells that inherited *Ddr*$^{dn}$ were significantly more likely to be in the trailer position, increasing the proportion of mCherry-labeled cells in the trailer position to 66% (*Figure 2C*). Mosaic expression of RhoDF$^{ca}$ has the opposite, albeit not statistically significant, effect, promoting positioning of the labeled cell anteriorly (*Figure 2C*), further lending support to our hypothesis of leader cells being more protrusive and less retractive than trailers.

Of note, one simulation of more acute reduction of cell-matrix adhesion in the trailer of leader-trailer polarized cell pairs occasionally caused in silico tumbling behavior (*Figure 2—figure supplement 1*), where the low-adhesion trailer cell climbs on top of and over the leader cell with normal adhesion. We previously observed this distinctive behavior in vivo (*Figure 2—figure supplement 1*), following TVC-specific inhibition of cell-matrix adhesion, and reduction of collagen9-a1 secretion from the adjacent trunk endoderm (*Gline et al., 2015*; *Bernadskaya et al., 2019*). Taken together, these observations indicate that reduced cell-matrix adhesion promotes positioning of the cell with reduced adhesion posterior to the cell with more adhesion, thereby preferentially adopting the trailer state, which in turn suggests that cell-matrix adhesion is stronger in leader cells.

Our previous work has suggested that leading cells are more adhesive than trailer cells (*Bernadskaya et al., 2019*). To evaluate the distribution of adhesion-associated actin structure, we assayed the distribution of F-actin in leader/trailer cell pairs using Lifeact::GFP. The F-actin marker localized to the leading edge and an intracellular punctate pattern, suggesting association with adhesion complexed and/or intracellular vesicles (*Figure 2D*). Quantifications indicated that leader cells consistently contained more ventral Lifeact::GFP+ F-actin puncta than trailer cells, supporting the prediction that F-actin-rich adhesion complexes are enriched in leading cells during TVC migration (*Figure 2D*). Taken together, these results indicate that increased cell-matrix adhesion and protrusive activity are hallmarks of the leader cell state during collective cell migration.

## Hierarchical guidance orients collective polarity in the direction of migration

From the above sections, a picture emerges whereby the supracellular organization of migrating pairs of cardiopharyngeal progenitors is characterized by leader-polarized protrusive activity and cell-matrix adhesion, and trailer-polarized deadhesion and myosin-driven retraction. Both experimental and simulated disruptions of this supracellular polarity alter directionality, marked by alignment of the leader-trailer axis with the direction of migration. However, the two cells are not arranged in a linear

leader-trailer orientation at birth (*Figure 3A*), and previous observations indicated that either cell can assume a leader position, although single-cell lineage tracing indicates that the leader emerges from the most anterior cell in ~95% of embryos (*Gline et al., 2015*).

We analyzed the establishment of leader/trailer polarity from the initial time of TVC birth to full polarization and alignment with the direction of migration over four embryonic stages encompassing TVC migration (*Hotta et al., 2007*). Tracking $cos(\theta)$ to quantify the alignment of cell pairs with direction of migration, with $\theta$ defined as the angle between the leader-trailer axis (axis connecting their centers of mass) and the direction of migration (*Figure 3A and B*), shows that prior to migration at embryonic stage 17 (8.5 hr post fertilization [hpf]), the leader/trailer axis is more orthogonal to the future direction of anterior movement. The cells reach their full polarization and alignment with direction of migration by stage 21, a process that takes approximately 1.5 hr at 18°C, after which they continue to migrate as a fully polarized cell pair for approximately 2 hr (*Figure 3A and C*).

We sought to explore possible cell-autonomous mechanisms governing the establishment of collective leader-trailer polarity and its alignment with the migration path. We modeled three possible directional modes for the cell pair without including directional noise (*Figure 3B*): the independent mode, where the two cells have the same distributions of retractive and protrusive forces and their retractive-protrusive axes are aligned to the right, along the net directional signal from the surrounding tissue; the faster-slower mode, where the total retractive-protrusive force in one cell is scaled up compared to the other cell, thus making one cell faster than the other; and the leader-trailer mode, where the prospective leader's retractive-protrusive axis is aligned with the external signal direction, while the prospective trailer's retractive-protrusive axis is oriented toward the leader's center of mass. In simulations, the two cells are initially placed side-by-side, with their retractive-protrusive axes orthogonal to the migration path (i.e., $cos(\theta) = 0$), thus resembling the arrangement observed in vivo prior to the onset of migration. Simulations show that the cells following the independent mode maintain their side-by-side orientation and fail to align with the migration path (*Figure 3DVideo 3*). By contrast, either the faster-slower or leader-trailer mode led cells to rearrange into a single file aligned with the migration path (i.e., $cos(\theta) = 1$), with the leader-trailer mode allowing faster alignment, with a half-time to alignment reduced compared to the fast-slower mode (*Figure 3D*, *Video 4*). This predicted behavior agrees qualitatively with the progressive polarization of TVCs observed in vivo.

The assumption that the basic rule of follower cells polarizing hierarchically towards leader cells predicts that cells should adopt a linear arrangement of leader and follower states even if the number of migrating cells was increased. To test this, we further probed the distinct migration modes by modeling three migrating cells (*Figure 3E*, diagram at top). In these simulations, each of the three cells adheres equally to the other two. The modes are similar to those for two cells (*Figure 3B*), with two adaptations: in the faster-slower mode, one cell is the fastest, another – the slowest, and the third moves with an intermediate speed; the leader-trailer mode becomes the leader-middle-trailer mode, in which the polarization axis of the leader is fixed to the external signal direction, the middle cell's axis orients towards the leader's center, and the trailer's axis orients towards the middle cell's center. To mirror the initial arrangement observed in vivo, we start three-cell simulations with individual cells distributed in a triangular pattern (*Figure 3E*, *Videos 5 and 6*). Similar to two-cell simulations, the independent mode failed to produce linear arrangements and the cells remained in triangular formation with no leader emerging, potentially due to the three-cell system minimizing the adhesive energy when each cell maintains contacts with the other two (*Figure 3E*). Although cells arranged more linearly under the faster-slower mode (*Figure 3E*, *Video 5*), they failed to align with the direction of migration for an extended period of time, only achieving the single-file arrangement towards the end of simulations (*Figure 3E*). The leader-middle-trailer mode was again the most effective at producing full polarization and linear order rapidly (*Figure 3E*, *Video 6*), suggesting that basic hierarchical rules of collective polarization can produce linear arrangements of cell groups containing variable numbers of cells. We tested this prediction in vivo, where ectopic FGF/M-Ras/Mek-driven induction within the *Mesp*+ lineage causes three or four cells to assume a cardiopharyngeal identity and migrate collectively (*Christiaen et al., 2008*; *Davidson et al., 2006*; *Razy-Krajka et al., 2018*). In conditions such as misexpression of a constitutively active form of M-Ras using the *Mesp* enhancer (*Razy-Krajka et al., 2018*) (*Mesp>M-Ras^{ca}*), the cells align in the direction of movement in 57% of the experimental embryos, with a single anterior leader, followed by two (13%) or three (87%) cells arranged in a single

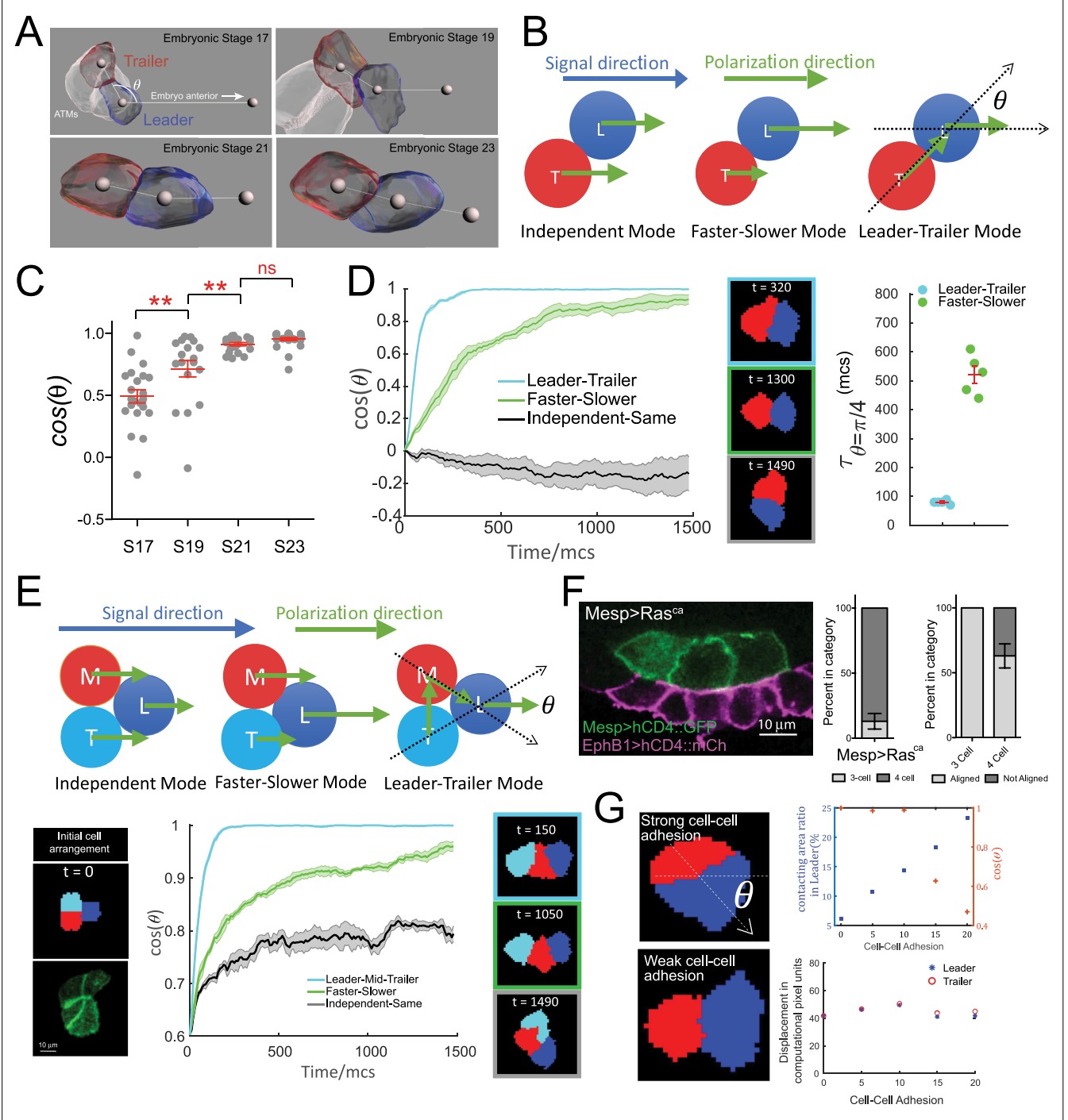

**Figure 3.** Hierarchical organization of multicellular migratory clusters. (**A**) Evolution of trunk ventral cell (TVC) polarization. Panels show in vivo-rendered images of cells at the indicated embryonic stages. Leader in blue, trailer in red, non-migratory anterior tail muscles (ATMs) in white. Spheres inside cells mark the center of mass, sphere to the right indicates direction of anterior migration. Angle theta between the axis of leader/trailer and direction of migration is indicated. (**B**) Three hypothesized polarization modes for two-cell migration. Independent: cells polarize independently in the signal direction and move with the same speed. Faster-slower: cells polarize independently in the signal direction, but one cell moves faster than the other. Leader-trailer: one cell (leader) follows the signal direction, while the other (trailer) polarizes in the direction of the leader's center of mass. L, leader; T, trailer. (**C**) Establishment of alignment between the leader/trailer axis and direction of migration. $Cos(\theta)$ is shown for indicated embryonic stages. Data were pooled from two biological replicates. Statistical analysis was performed using one-way ANOVA and Bonferroni post test. **p<0.01. (**D**) Left: the simulated evolution of two cells' geometry, quantified as cosine of the angle between the line connecting the cells centroids and the signal direction, ($cos\theta$), for the three polarization modes shown in (**B**). Five simulations are run for each mode, and the shaded area shows the standard error. Center:

*Figure 3 continued on next page*

*Figure 3 continued*

representative snapshots of two cells reaching linear arrangement or at the end of simulation using each mode. The colors of the frames correspond to the dataset on the graph. Right: scatter plot showing the time when $\theta$ reaches $\pi/4$ for two modes with mean and standard error and statistical analysis using Student's *t*-test with Welch's correction. (**E**) Top: hypothesized polarization modes for three cells. Independent: cells polarize independently in the signal direction and move with equal speeds. Faster-slower: cells polarize independently in the signal direction, in this case, the leader travels the fastest, trailer the slowest, and middle cell travels at an intermediate speed. Leader-trailer: one cell (leader) follows the signal direction, middle cell polarizes towards the leader, and trailer polarizes toward the middle cell. Bottom: simulation of the three-cell group polarization under the three hypothesized polarization modes. Left: the initial cell arrangement in silico (top) and in vivo (bottom). Note that in vivo there are always four cells prior to migration arranged in a rectangular pattern. Center: the polarization of migrating cell clusters over time is quantified by the cosine of the angle between the lines connecting the leader and the two posterior cells separately, as shown in (**F**). Five simulations were run for each mode, and shaded area shows the standard error. Right: representative snapshots when the three cells reach linear arrangement for the three modes examined (for the independent-same mode, linear arrangement is never reached, snapshot shows cells at the end of a simulation run). The colors of the frames correspond to the datasets on the graph. (**F**) Three migratory cells are linearly arranged in the direction of migration in vivo. Bar graphs show the proportion of TVCs that migrate as either three or four cells under induced MAPK signaling by *Mesp>Ras^{ca}* and proportion of cell groups that are linearly polarized in each subset. Data were pooled from two biological replicates. Error bars show standard error of proportion. (**G**) Effects of modulating cell-cell adhesion on the contacting area between the two cells and on their speed, quantified by the percentage of total surface area of the leader cell (top graph, left y-axis, blue symbols), on the ability of the cell pair to polarize in the direction of migration quantified by $cos\theta$ (top graph, right y-axis, red symbols), where $\theta$ is the angle between the line connecting two cells and the moving direction as shown in the top image on the left, and on the total displacement of the leader/trailer pair (bottom graph). x-axis shows the relative energy of the cell-cell junction (the adhesion parameter is rescaled here so that larger value means stronger cell-cell adhesion). Images show cell pairs with either high (top) or low (bottom) cell-cell adhesion. Arrow represents leader/trailer axis.

line (68%, n = 31) (*Figure 3F*, *Videos 7 and 8*). This provides experimental support for our leader-trailer polarization model.

Since the ability to organize cell collectives requires maintenance of cell-cell junctions, we also investigated the effect of cell-cell adhesion strength on collective polarization. Multiple simulations varying the cell-cell adhesion energy in our basic model show that cell-cell adhesion strength does not affect the total displacement of cell pairs over long time intervals (*Figure 3G*, bottom right). However, increasing or decreasing cell-cell adhesion energy causes the cell-cell boundary area to increase or decrease, respectively (*Figure 3G*, top right, blue plot), which leads to a drastic misalignment of cells with the direction of migration beyond a cell-cell adhesion strength of 10 (relative energy units). This suggests that the extent of cell-cell adhesion may be regulated in vivo and that cells with high cell-cell adhesion will reorient their supracellular polarity away from the direction of migration.

## Collective polarity fosters persistent directionality

Having characterized ground rules that govern collective arrangement of migrating cardiopharyngeal progenitors, we sought to explore the specific properties conferred by this supracellular organization. Unlike specialized motile cells, such as *Dictyostelium*, fish keratocytes, or neutrophils, which can migrate at ~10 µm/min, 7.5 µm/min, and ~19 µm/min, respectively (*Buenemann et al., 2010*; *Graham et al., 2013*; *Hoang et al., 2013*), TVCs move at ~0.4 µm/min, which is relatively slow, but not unexpected for a developmental migration that contributes to the establishment of accurate cellular patterns in the embryo (*Trepat et al., 2012*). We reasoned that this behavioral accuracy would be reflected in the cells' persistence, defined as the ratio of beginning-to-end displacement to trajectory length (*Gorelik and Gautreau, 2014*; *Figure 4A*). Directional noise, which counters persistence and accuracy, emerges from inherent stochasticity of motile engines, random fluctuations of external cues, and/or signal transduction (*Tang et al., 2014*) and can cause meandering trajectories of cells (*Wu and*

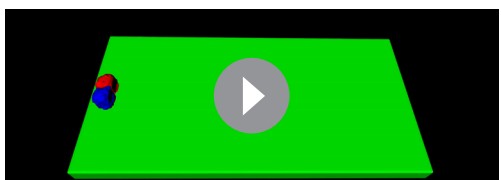

**Video 3.** In silico modeling of cell position rearrangement of two migrating cells using the faster-slower mode of migration.

https://elifesciences.org/articles/70977/figures#video3

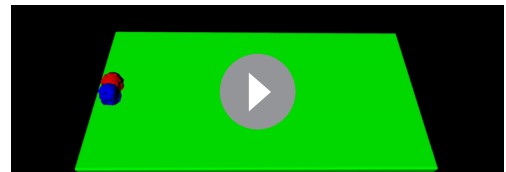

**Video 4.** In silico modeling of cell position rearrangement of two migrating cells using the leader-trailer mode of migration.

https://elifesciences.org/articles/70977/figures#video4

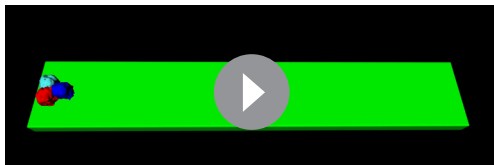

**Video 5.** In silico modeling of cell position rearrangement of three migrating cells using the faster-slower mode of migration.

https://elifesciences.org/articles/70977/figures#video5

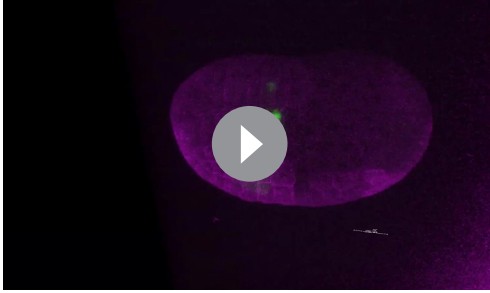

**Video 7.** In vivo conversion of anterior tail muscle (ATM) fate to trunk ventral cell (TVC) by misexpression of *Mesp>Ras*^ca, resulting in migration of four cells. B7.5 lineage. Nuclei are marked with *Mesp>H2B::GFP*. Epidermal cells are marked with *EphB1>hCD4::mCherry*. Epidermal marker is used to orient the embryo.

https://elifesciences.org/articles/70977/figures#video7

*Zhang, 2015*). We therefore simulate noise by adding directional stochasticity to the model of two cells in the leader-trailer mode and compare the persistence of migrating cell pairs with that of single cells (*Figure 4A and B*). In simulations, motile cell pairs are always more persistent than single cells when centers of mass are tracked over time for each condition, suggesting that single cells are more sensitive to directional stochasticity. Of note, the total length of the migration path in the simulations was not altered, suggesting that the decreased displacement is a function of the meandering path traveled by the less persistent single cells (*Figure 4B*).

We compared the final displacement of single TVCs in vivo to the total displacement of the leader TVC. In agreement with simulations, TVC pairs migrated further from the anterior ATM than single cells (*Gline et al., 2015*; *Figure 4C*). To test if this was due to the loss of persistence, as predicted by the above simulations, we compared the persistence of cell pairs by tracking the nuclei of leader and trailer cells in 4D datasets and compared to the migration paths of single TVCs (*Figure 4D*, *Videos 9 and 10*). In control conditions, the leader and trailer migrate with similar persistence; however, the migration paths of single TVCs were significantly less persistent than those of either leader or trailer cells, suggesting that in vivo, collective organization confers robust directionality to the migrating cells.

In summary, both simulations and in vivo observations suggested that polarized cell pairs migrate with increased robustness to fluctuations in directionality compared to single cells.

## Polarized cell pairs overcome mechanical resistance from the endoderm during migration

The above sections indicate that collective organization endows the TVCs with defined properties (e.g., persistence) that are intrinsic to cell pairs and determine the characteristics of their migration. However, TVCs migrate surrounded by embryonic tissues that canalize their behavior (*Christiaen et al., 2008*; *Gline et al., 2015*; *Bernadskaya et al., 2019*). Specifically, shortly after the onset of migration, the TVCs penetrate the extracellular

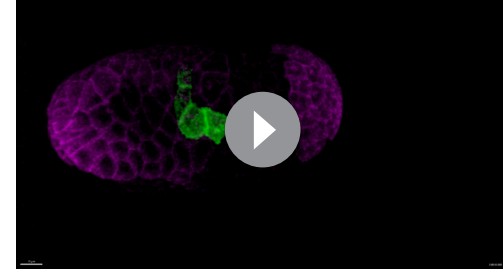

**Video 8.** In vivo conversion of anterior tail muscle (ATM) fate to trunk ventral cell (TVC) by misexpression of *Mesp>Ras*^ca, resulting in migration of four cells. B7.5 lineage. Cell membranes are marked with *Mesp>hCD4::GFP*. Epidermal cells are marked with *EphB1>hCD4::mCherry*. Epidermal marker is used to orient the embryo.

https://elifesciences.org/articles/70977/figures#video8

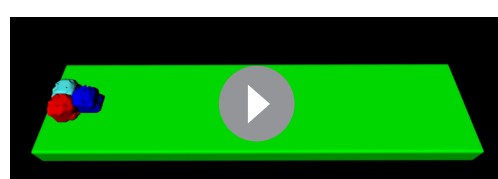

**Video 6.** In silico modeling of cell position rearrangement of three migrating cells using the leader-trailer mode of migration.

https://elifesciences.org/articles/70977/figures#video6

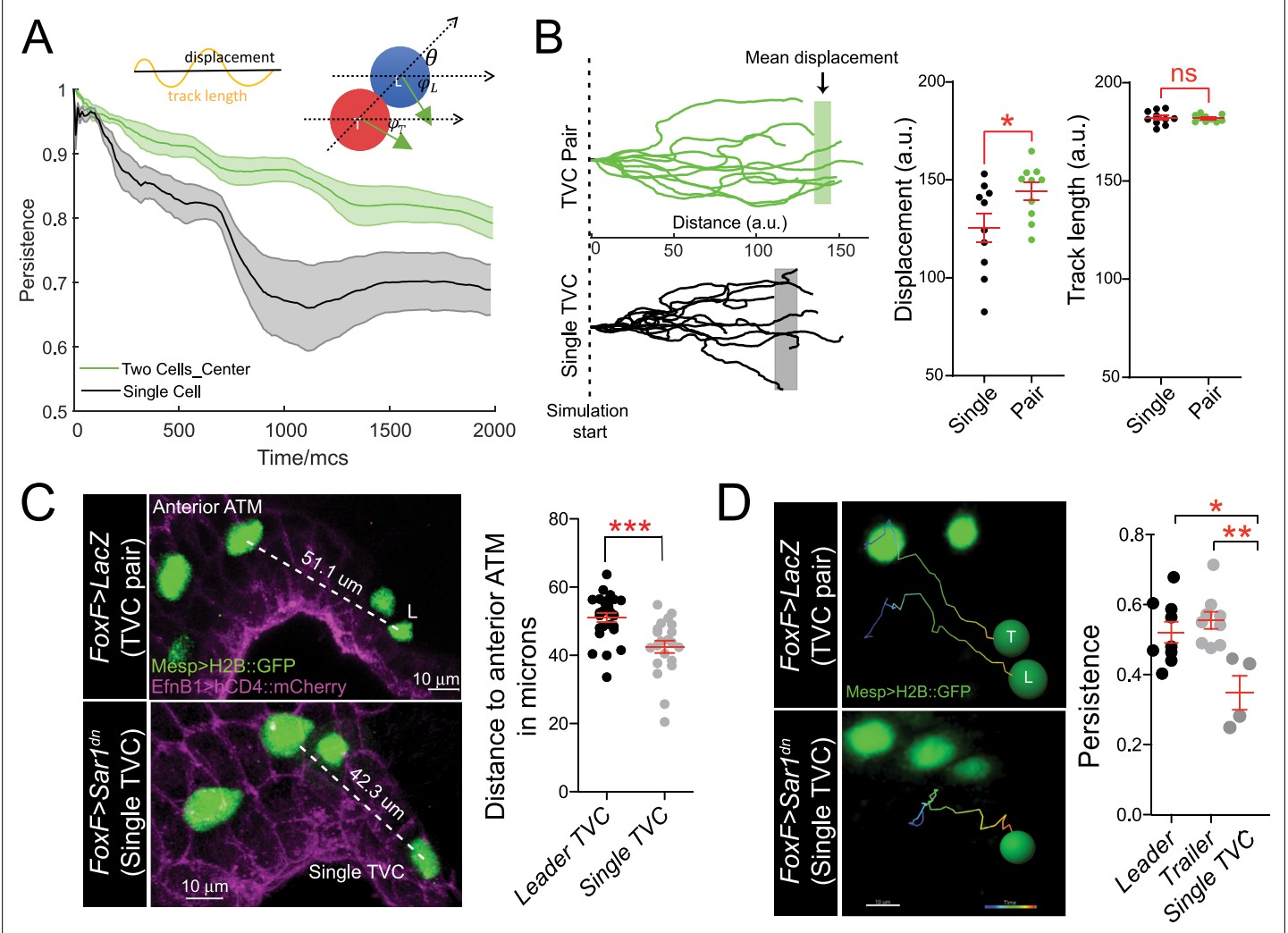

**Figure 4.** Migratory persistence of cell pairs and single cells. (**A**) Left: simulation of migration persistence over time for single cell and the centroid of the cell pair. The shaded area shows the standard error. In the model, green arrows show the direction of active forces for each cell that is directionally biased but also fluctuates randomly. The leader cell is biased to the right, which is the direction of the external cue, and the trailer cell is biased to the leader cell. $\varphi_L$ and $\varphi_T$ are the angles between the green arrows and the right direction, and $\theta$ is the angle between the line connecting two cells and the right direction. The specific stochastic equation for $\varphi$ is given in the Materials and methods section using the same angle notations. The diagram on top shows the relationship between displacement and track length and $persistence = \frac{displacement}{track\ length}$. (**B**) Comparison of tracks from simulations of either migrating cell pairs or single cells within the same simulation time. Shaded vertical lines represent mean final displacement. Graphs on the right show mean total displacement and mean total track length with standard error. Statistical analysis was performed using Student's $t$-test. *p<0.05. (**C**) In vivo analysis of total displacement of leader cells in a cell pair and single trunk ventral cells (TVCs) from the anterior tail muscle (ATM). TVC and ATM nuclei are marked with *Mesp>H2B::GFP*, epidermal cell membranes are marked with *EphB1>hCD4::mCherry*. Data were pooled from two biological replicates. Scatter plot shows average displacement and standard error. ***p<0.001. (**D**) In vivo migration of TVC pairs compared to single TVC. Nuclei of the cells are used to track cell migration path in 4D datasets. Paths are color-coded from early (blue) to late (red). Scatter plot shows mean persistence of leader (n = 8), trailer (n = 8), and single TVC (n = 4) with standard error. Statistical analysis was performed using one-way ANOVA with Bonferroni post test. *p<0.05, **p<0.01.

space between the ventral trunk epidermis, which they use as stiff substrate, and the softer trunk endoderm, which locally deforms as TVCs progress anteriorly (*Gline et al., 2015*; *Figure 5A*). We reasoned that the trunk endoderm likely exerts mechanical resistance to the passage of the TVCs and hypothesized that this resistance may be better overcome by polarized cell pairs (*Figure 5A and B*). We predicted that when two cells migrate in a leader-trailer arrangement, the surface that the leader exposes to mechanical resistance is equivalent to that of a single cell, while the trailer pushes from the rear, therefore adding forward-bearing compression force to overcome endoderm resistance (*Figure 5B*).

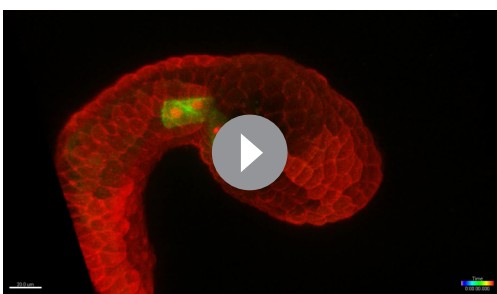

**Video 9.** In vivo migration of a control trunk ventral cell (TVC) pair. Nuclei are marked with *Mesp>H2B::GFP*. Epidermal cells are marked with *EphB1>hCD4::mCherry*. Epidermal marker is used to orient the embryo. Track traces the path of the nucleus centroid during migration.
https://elifesciences.org/articles/70977/figures#video9

To explore this argument, we added deformable endoderm cells to our model and simulated cell migration under varying endoderm stiffness, comparing the effects on migration speed. We simulate migration of one- and two-cell systems, tracking the simulated cells' centers of mass with five simulations run per condition shown in *Figure 5D*. When modeling a softer endoderm, by modulating volume preservation and cortical tension parameters relative to those of the TVCs (see Appendix 1), polarized cell pairs perform best, whether in supracellular mode or equivalent force distribution, while side-by-side cells were the slowest, presumably because they expose a greater surface to mechanical resistance (*Figure 5B and D*). Notably, two equivalent cells moving in single file advanced marginally faster than the supracellular pair, suggesting that cell alignment itself is a key determinant of efficient

migration under the soft tissue. When simulating a stiffer endoderm, the advantage of supracellular organization became more apparent. In these simulations, the supracellular collective migrated faster than other arrangements (*Figure 5D*). Notably, cells migrating side-by-side were slower than single cells with either endoderm stiffness (*Figure 5D*), which is consistent with the notion that a more extended surface of contact with the endoderm exposes them to greater mechanical resistance, while the smooth teardrop shape of single cells may be near optimal for lowering the resistive deformations of the endoderm. This suggests that the collective shape resulting from supracellular organization optimally minimizes mechanical resistance of the surrounding tissue to migration.

The predicted relative speeds of cells migrating under the endoderm are based on the simplest model, which assumes that the endoderm's primary effect is mechanical resistance to deformation. In vivo, the interactions between TVCs and endoderm cells are likely more complex, involving indirect cell-cell signaling via extracellular deposition of collagen9-a1 by the endoderm (*Bernadskaya et al., 2019*). In summary, the combined in silico simulations and in vivo observations indicate that the collective organization of migrating cardiopharyngeal progenitors allows them to overcome mechanical resistance from the deforming endoderm and reach a typically mesodermal position between germ layers for cardiac organogenesis.

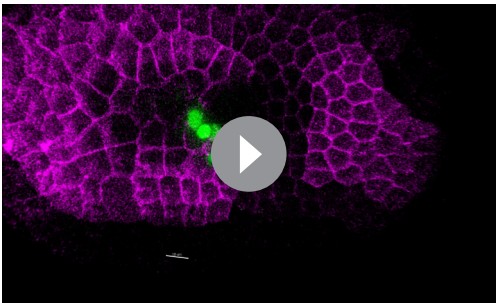

**Video 10.** In vivo migration of a single trunk ventral cell (TVC), produced by misexpression of *Foxf >Sar1dn*. Nuclei are marked with *Mesp>H2B::mCherry*, and cell membranes are marked with *Mesp>hCD4::GFP*. Epidermal cells are marked with *EphB1>hCD4::mCherry*. Epidermal marker is used to orient the embryo. Track traces the path of the nucleus centroid during migration.
https://elifesciences.org/articles/70977/figures#video10

## Discussion

Complex multicellular behaviors, including directed collective cell migration, emerge from the context-specific integration of universal dynamic processes, which operate at subcellular scale and are coordinated within and across cells (*Bernadskaya and Christiaen, 2016*). The sheer complexity of integrated cellular systems constrains direct experimental interrogations, but mathematical models and simulations provide a powerful complement to probe the relative biophysical contributions of subcellular processes to cellular behavior.

In this study, we used a mathematical model, built from first biophysical principles, to generate computational simulations and explore the morphodynamic space of motile cardiopharyngeal progenitor cell pairs of the tunicate *Ciona*. Qualitative comparisons with experimental data

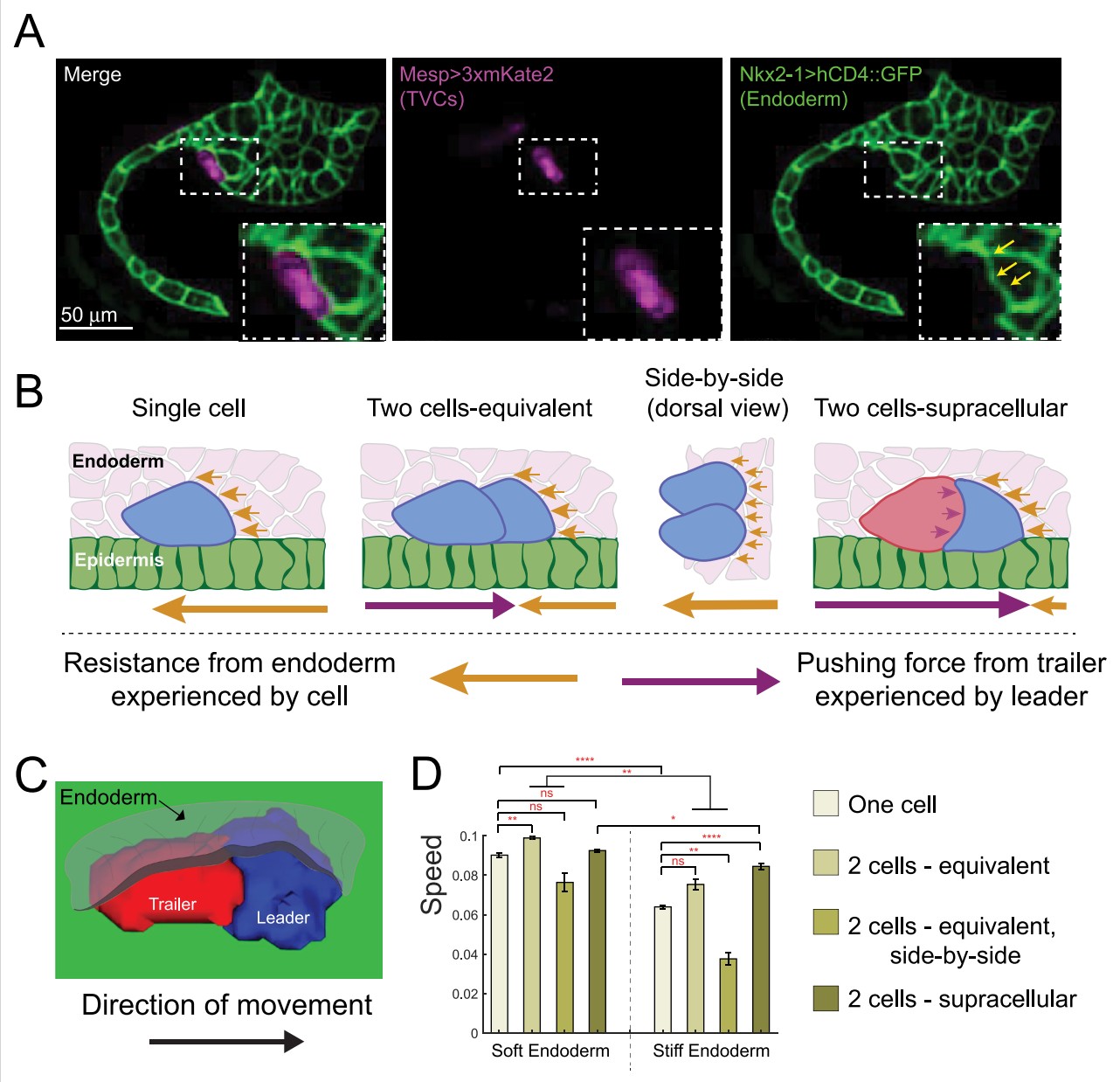

**Figure 5.** Supracellular cell pairs are more efficient at dispersing pressure from surrounding tissues. (**A**) Micrographs of stage 23 embryos showing the endodermal pocket formed during trunk ventral cell (TVC) migration. Embryos are oriented with anterior to the right. Endodermal cells are marked with *Nkx2-1>hCD4::GFP* (green), TVCs are marked with *Mesp>3xmKate2* (magenta). Yellow arrows point to depression pocket left in the endoderm by migrating TVCs. (**B**) Proposed model for higher efficiency of supracellular cell pairs in overcoming resistance from the endodermal tissue (pink) during migration: the adhesive cell pair shares the resistance force (yellow arrows), which otherwise each single cell must overcome alone. Pressure from the posterior trailer (purple arrows) can help the cell pair overcome resistance from the endoderm. Size of the arrows below the graphics represents relative strength of the force experienced by the cell in the direction of the arrow. (**C**) Simulated supracellular cell pair underneath the endoderm. Epidermis is shown in green. The endoderm is rendered transparent. (**D**) Speed comparison between single cell and differently arranged cell pairs with different profiles and force distributions under the endoderm of varying stiffness. Five simulations are run for each condition; the error bar is the standard error. Statistical analysis is performed using Brown–Forsythe and Welch ANOVA test. *$p<0.05$, **$p<0.01$, ****$p<0.0001$.

indicated that the shape of cell pairs, similar to that of a single motile cell, emerges from the distribution of higher protrusive activity and cell-matrix adhesion to the leader cell, whereas the rear of the trailer cell is the primary site of myosin-based retraction. The latter prediction is corroborated by in situ patterns of myosin activity and F-actin distribution. This illustrates that a purely mechanical model such as CPM, which assumes that most active stresses are generated at the cellular periphery

in addition to hydrostatic pressure in the cytoplasm, can uncover the biomechanical underpinnings of collective cell shape and movement.

The above patterns of protrusive activity, cell-matrix adhesion, and contractility might seem trivial, considering the well-established organization of individual migrating cells. However, in the 'supracell,' the distribution of various cytoskeletal activities across all cells in a collective suggests the existence of mechanisms to ensure such 'division of labor.' This simple prediction implies multiple roles for the cell-cell contact, in addition to its anticipated low surface tension.

First and foremost, cell-cell adhesion must be strong enough to maintain the integrity of the collective and permit mechanical coupling, lest cells lose contact and migrate disjointly (*Figure 1—figure supplement 1C*). Conversely, the model predicts that excessive cell-cell adhesion antagonizes cell-matrix adhesion and disrupts collective polarity. Balancing cell-cell and cell-matrix adhesion may result from either mechanical interaction, as suggested by the model, and/or biochemical cross-talks, as observed in other systems (*Martinez-Rico et al., 2010*; *Ramprasad et al., 2007*). The coexistence and contributions of both cell-cell and cell-matrix adhesion to supracellular migration emphasize the hybrid nature of such multicellular systems, where cells adopt intermediate states on an epithelial-to-mesenchymal continuum (*Friedl and Mayor, 2017*; *Bernadskaya and Christiaen, 2016*; *Lecaudey and Gilmour, 2006*).

Close cellular contacts probably facilitate propagation of direct mechanical and biochemical interactions that underlie supracellular migration. We tentatively distinguish 'information flows' that propagate in either a back-to-front or a front-to-back fashion (*Capuana et al., 2020*; *Mayor and Etienne-Manneville, 2016*). For instance, similar to *Xenopus* cranial neural crest cells and the zebrafish lateral line primordium (*Yamaguchi, 2021*), *Ciona* cardiopharyngeal progenitors appear to focus contractility at the back of the trailer cell, thus suggesting that a 'rear-wheel' engine may help power their migration. However, in contrast to neural crest cells and *Drosophila* border cells, we did not observe supracellular actomyosin 'cables' extending across cardiopharyngeal progenitors and there is no exchange of cell positions between TVCs. Instead, we surmise that this 'rear-wheel' drive represents a back-to-front mechanical input, which propagates as compression force and emerges from rear-localized myosin activity, possibly in response to chemorepulsive inputs integrated by the trailer. It is also conceivable that cell-cell adhesion complexes suppress myosin-based contractility at the back of the leader cell, for example, through recruitment of Rho GAP molecules by cadherin, as is the case in early *Caenorhabditis elegans* embryos (*Klompstra et al., 2015*). In neural crest cells, CIL provides such back-to-front signals that polarize the cell collective, in part through cadherins, ephrin receptors, and planar cell polarity (PCP) pathway molecules (*Mayor and Etienne-Manneville, 2016*). The PCP pathway offers a particularly tantalizing explanation for the spontaneous alignment of three or four adhering cells following ectopic induction of the cardiopharyngeal progenitor fate, and the existence of a 'leader-trailer' mode of collective arrangement predicted by the model.

TVCs' collective polarity is marked by higher protrusive activity and cell-matrix adhesion in the leader, as indicated by experimental observations and model predictions (*Christiaen et al., 2008*). Mechanically, it is likely that adhesion complexes are established following lamellipodia formation and support traction forces, which complement rear-driven compression to propel the cells forward. It is conceivable that lower protrusive activity in the trailer limits the deployment of cell-matrix adhesion complexes. However, cell-matrix adhesion in the trailer is probably needed to anchor the cell and allow for its hydrostatic pressure to push the leader. Therefore, one must invoke mechanisms whereby the leader suppresses protrusive activity in the trailer, while permitting the establishment of trailer cell-matrix adhesion in its path. In *Drosophila* border cells, leader-driven suppression of protrusive activity in follower cells is mediated by Rac (*Cai et al., 2014*) and by Delta-Notch signaling between tip and stalk cells during angiogenesis (*Aspalter et al., 2015*). It is thus likely that similar 'front-to-back' mechanisms govern the collective distribution of protrusions, and by extension cell-matrix adhesion complexes, in cardiopharyngeal progenitors.

While future work combining biophysical modeling, force measurements, and/or inference from quantified cell shapes is needed to elucidate the mechanisms underlying supracellular organization in vivo, our model and experimental investigations uncover important consequences for directed migration: namely, persistence and mechanical interaction with surrounding tissues. Specifically, both computational simulations and live imaging indicated that the instantaneous directionality of single cells fluctuates more than that of cell pairs. In other words, polarized cell pairs are more persistent.

It is possible that cell pairs are better at buffering the noise inherent to navigating a complex and changing environment, in part by distributing interactions over greater surfaces, and integrating guidance cues more accurately.

Finally, our observations indicate that supracellular organization determines the outcome of interactions with surrounding tissues during migration. We previously determined that TVC pairs migrate onto the extracellular matrix (ECM) associated with the basal lamina of the ventral trunk epidermis (*Bernadskaya et al., 2019*), which presumably offers a stiff substrate permitting traction forces. Of note, a specific collagen, *col9-a1*, secreted from the trunk endoderm is deposited onto the ECM and necessary for TVC-matrix adhesion and collective polarity (*Bernadskaya et al., 2019*). Here, we find that the trunk endoderm resists deformation by migrating TVCs, which can nonetheless move forward by aligning and joining forces to push against and deform endodermal cells to penetrate the extracellular space. Our combined simulations and experimental observations thus suggest an effect of supracellular organization on the inter-tissue balance of forces that determine morphogenesis in the embryo.

## Resource availability

### Lead contact
Further information and requests for resources and reagents should be directed to and will be fulfilled by the lead contact: Yelena Bernadskaya (yb372@nyu.edu).

### Data and code availability
The codes generated during this study are available on GitHub (*Yue, 2021*; https://github.com/HaicenYue/3D-simulation-of-TVCs.git copy archived at swh:1:rev:c8a79bc7822f295ab72edb8e3f7660c823c3699e).

### Experimental model and subject details
Wild caught *Ciona robusta* (formerly *Ciona intestinalis* type A) were purchased from Marine Research and Educational Products (M-REP, San Diego, CA). As invertebrate chordates, animal care approval was not needed. Prior to use, animals were housed in a recirculating artificial seawater aquarium under constant illumination to prevent spawning.

# Materials and methods

**Key resources table**

| Reagent type (species) or resource | Designation | Source or reference | Identifiers | Additional information |
|---|---|---|---|---|
| Software, algorithm | | https://github.com/HaicenYue/3D-simulation-of-TVCs.git | | |
| Genetic reagent (*Ciona robusta*) | Wild-caught | M-Rep, San Diego,CA | https://www.m-rep.com | |
| Sequence-based reagent | RhoDFca-F | This paper | PCR primers | TGAAACTTGTATTGCGGCCGC |
| Sequence-based reagent | RhoDFca-R | This paper | PCR primers | agacgtacgt GAATTCTCACAATAGC AAACAACAGCAGCAG |
| Sequence-based reagent | iMyo::GFP – F | This paper | PCR primers | ACTTGTATTG CGGCCGCAACCAT GGCCGAGGTGCAGC |
| Sequence-based reagent | iMyo::GFP – R | This paper | PCR Primers | gctgagcgcGAA TTCTTACTTGT ACAGCTCGTCCATGC |
| Recombinant DNA reagent | pCESA: Mesp > hCD4::GFP (plasmid) | PMID:30610187 | | B7.5 lineage specific GFP membrane marker |

*Continued on next page*

*Continued*

| Reagent type (species) or resource | Designation | Source or reference | Identifiers | Additional information |
|---|---|---|---|---|
| Recombinant DNA reagent | pCESA: Mesp > H2B::GFP (plasmid) | PMID:30610187 | | B7.5 lineage specific GFP histone/nuclear marker |
| Recombinant DNA reagent | pCESA: Mesp > iMyo::GFP (plasmid) | This paper | | B7.5 lineage specific GFP myosin intrabody |
| Recombinant DNA reagent | pCESA: Foxf > mCherry (plasmid) | PMID:30610187 | | mCherry TVC-specific marker |
| Recombinant DNA reagent | pCESA: EfnB > hCD4::mCherry (plasmid) | PMID:30610187 | | Epidermal mCherry membrane marker |
| Recombinant DNA reagent | pCESA: Mesp > 3xmKate2 (plasmid) | PMID:30610187 | | B7.5 lineage specific mKate2 marker |
| Recombinant DNA reagent | pCESA: Nkx2–1> hCD4::GFP (plasmid) | PMID:30610187 | | Endoderm specific GFP cell membrane marker |
| Recombinant DNA reagent | pCESA: Foxf>Sar1$^{dn}$ (plasmid) | PMID:25564651 | | TVC-specific dominant negative Sar1 |
| Recombinant DNA reagent | pCESA: Foxf > Rhodf$^{ca}$ (plasmid) | PMID:18535245 | | TVC-specific constitutively active RhoD/F |
| Recombinant DNA reagent | pCESA: Mesp > LacZ (plasmid) | PMID:30610187 | | B7.5 lineage specific LacZ loading control |
| Recombinant DNA reagent | pCESA: Foxf > Intβ1$^{dn}$ (plasmid) | PMID:30610187 | | TVC-specific dominant negative Intβ1 |
| Recombinant DNA reagent | pCESA: Foxf > Ras$^{ca}$ (plasmid) | PMID:18535245 | | TVC-Specific constituitivley active Ras |
| Recombinant DNA reagent | pCESA: Foxf > Ddr$^{dn}$ (plasmid) | PMID:30610187 | | TVC-specific dominant negative Ddr |
| Software, algorithm | FIJI | *Schindelin et al., 2012* PMID:22743772 | *RRID*:SCR_002285 | |
| Software, algorithm | Bitplane Imaris | Bitplane Imaris | *RRID*:SCR_007370 | |
| Software, algorithm | Prism 9 | https://www.graphpad.com/ | RRID:SCR_002798 | |

## Electroporation and transgene expression

*C. robusta* (formerly known as *C. intestinalis* type A) adults were purchased from M-Rep. Gamete isolation, fertilization, dechorionation, and embryo incubation were performed as previously published (*Christiaen et al., 2009a*; *Christiaen et al., 2009b*). The amount of DNA electroporated varied from 10 µg to 90 µg. Animals were reared at 22–24°C. Embryos used for direct visualization of fluorescent markers were fixed in 4% MEM-FA for 30 min, cleared with an PBST-NH$_4$Cl solution (50 mM NH$_4$Cl, 0.15% Triton-X100, 0.05% Tween-20 in 1× PBS), mounted in 50% glycerol supplemented with 2% Dabco 33-LV antifade reagent (Sigma-Aldrich, #290734) and imaged using a Leica SP8 X Confocal microscope.

## Live imaging and TVC tracking

To generate 4D datasets, embryos at 4.5 hpf FABA stage 15 were mounted on glass-bottom microwell Petri dishes (MatTek, part# P35G-1.5–20C) in artificial seawater. Plates were sealed by piping a border of vaseline and 5% (v/v) mineral oil (Sigma, #M841-100ml) and covered with a 22 × 22 Fisherbrand Cover Glass (#12-541-B). Embryos were imaged on a Leica inverted SP8 X Confocal microscope using the 40× water immersion lens at 512 × 512 resolution every 3.5 min for 4–5 hr. B7.5 lineage nuclei and epidermal cell membranes were visualized using *Mesp >H2B::GFP* and *EfnB>hCD4::mCherry*, respectively, and TVC migration was tracked using Bitplane Imaris Software Spots module.

## Image acquisition

All images were acquired using the Leica SP8 X WLL confocal microscope using the 63× glycerol immersion lens, NA = 1.44. Z-stacks of fixed embryos were acquired at the system optimized Z-step,

512 × 512 resolution, 600 Hz, and bidirectional scanning. Multiple HyD detectors were used to capture images at various wavelengths.

## Quantification and statistical analysis

### Morphometrics analysis

The membrane marker *Mesp>hCD4::GFP* was used to segment the TVCs and derive morphometric measurements such as sphericity, area, and volume in Bitplane Imaris using the Cell function with cell segmentation calculated from cell membranes with an average cell size of 6. Thresholding is adjusted based on individual image properties. Z-steps were normalized to achieve equal voxel size in X, Y, and Z planes. TVCs were then segmented and resulting cells were exported to separate surfaces. For experiments described in *Figure 2A*, no marker was used to follow transmission of *Foxf>Ddr^{nd}* and all cells were analyzed. Under these conditions, there is an 80% chance that any given cell has inherited the Ddr^{dn} perturbation. To calculate the distance or angle between cells, a point was placed at the center of mass for each cell using either the nucleus or the cell object using the Bitplane Imaris Measurement module.

### Image preprocessing

All 3D stacks were imported and converted to Imaris format. Images were reoriented and cropped with the leader cell to the left. An automated Gaussian filter and background subtraction was applied to all images using the Imaris Batch module. Images were projected to 2D using Maximum Intensity Projection to reflect the dorsal view of the cells and exported as TIFFs for analysis in Fiji.

### Aspect ratio calculation

2D projected images were imported into Fiji and converted to 8-bit format. A threshold was applied to each individual image, and empty spaces were filled using the Binary -> Fill Holes function. Resulting object was used to derive the aspect ratio using the Analyze Particles function. In simulation, black-and-white images were obtained using imbinarize function in MATLAB and then regionprop.BoundingBox function was used to get the minimal rectangle that enclose the object. Aspect ratio is the ratio between the width and length of this rectangle.

### Myosin intensity analysis

Images were imported into Fiji. Using the freehand line function with a width of 10 units, a scan was performed on the leading edge of the leader cell, the cell-cell junction, and the membrane of the trailer cell using the membrane marker as a guide. The intensity of iMyoGFP and the membrane marker *Mesp>hCD4::mCherry* along the scan was measured using the Plot Profile function and exported as intensity along the line scan. This was done for each cell pair. Readings along the line scan were aligned based on the starting position of the scan and averages were calculated.

### Lifeact::GFP distribution analysis

The Cell function was used to segment leader and trailer cells with vesicle detection. The membrane marker *Mesp>hCD4::mCherry* was used for membrane detection based on membrane signal intensity. Vesicles or spots were detected in each cell object, with the volume growing option to allow for detection of vesicles of varying size. The threshold setting for spot detection in the green channel was set to use the top 10% of GFP signal intensity.

### Statistical analysis and data representation

For all data comparing two samples of continuous variables, the Wilcoxon rank-sum test (also known as the Mann–Whitney test) was used. Categorical data were analyzed using Fisher's exact test. Simulation data were analyzed using Student's *t*-test with Welch's correction. For datasets containing more than two conditions and taking into account cell type (leader/trailer), a two-way ANOVA followed by the Bonferroni post test was used. For all datasets containing nominal variables, a chi-square test was used. p-Values are reported as follows: *p<0.05, **p<0.01, ***p<0.001.

## Model

We use the CPM (*Graner and Glazier, 1992*) to simulate the movement of one or several cells on the substrate. The model is based on the minimization of the effective energy *H*, which is a function of the cell shape and areas of contact between adjacent cells and between cells and substrate. It is computationally efficient to study the multiple 3D cells with enough resolution. The model also allows adding protrusive and retractive forces to the cells (*Rens and Edelstein-Keshet, 2019*; *Li and Lowengrub, 2014*; *Szabó and Merks, 2013*).

In the CPM, the space is divided into pixels (in the model, the cell size is about $10 * 10 * 10$ pixels), and each pixel   is assigned a spin $\sigma_i$. The spin is effectively an index that identifies which cell the pixel belongs to. A stochastic modified Metropolis algorithm (*Cipra, 2018*) was used to determine how the spin $\sigma$ changes. At each step, the algorithm randomly selects a target site,  , and a neighboring source site $j$. If they belong to different cells, or to a cell and neighboring environment, the algorithm sets $\sigma_i = \sigma_j$ with probability, $P_{\sigma_i \rightarrow \sigma_j}$, which is determined by the Boltzmann acceptance function:

$$P_{\sigma_i \rightarrow \sigma_j} = \left\{ 1, \ \Delta H \leq 0 \ e^{-\frac{\Delta H}{T}}, \ \Delta H > 0 \right.$$

where $\Delta H$ is the change of the effective energy caused by this change of spin, and $T$ is an effective temperature parameter describing the amplitude of stochastic fluctuations of the cell boundary (*Swat, 2012*). We use $T = 10$ for all the simulations in this paper. The key part of any specific CPM is the effective energy $H$. In our model, we define $H$ as

$$H = \sum_\sigma \lambda_\sigma \left( v_\sigma - V_\sigma \right)^2 + \sum_\sigma \kappa_\sigma a_\sigma^2 \quad + \sum_{\sigma_1, \sigma_2} J_{\sigma_1 \sigma_2} S_{\sigma_1 \sigma_2} + \sum_i \left( W_{\sigma_i, p} \left( \vec{r}_i \right) + W_{\sigma_i, r} \left( \vec{r}_i \right) \right),$$

Here, the first and the second terms represent the effects of the volume conservation and cell surface (cortex) contraction, respectively. Alternatively, the first term can be thought of as the effect of the hydrostatic pressure of the cytoplasm, and the second term – as the effect of the cell cortex tension. The third term represents the adhesion energy between the neighboring cells and between the cells and the ECM (also called substrate or ECM below). The last term is the effective potential energy related to the protrusive and retractive forces (with subscript $p$ and $r$, respectively).   is the pixel's index, and $\sigma$ is the cell's or environment's spin. Variables $v_\sigma$ and $a_\sigma$ are the volume and surface area of the cell $\sigma$, and $V_\sigma$ is its target volume. Unlike in some variants of the CPM, we keep the target surface area equal to zero, so effectively the cortex is contractile for any area. The target volume is a parameter that we take to be equal to the volume of the cube of the characteristic cell size. Parameters $\lambda$ and $\kappa$ are the coefficients determining how tightly the volume is conserved and how great the cortex tension is, respectively. Parameter $J_{\sigma_1 \sigma_2}$ is the adhesion energy per unit area of the boundary between cells $\sigma_1$ and $\sigma_2$ (or between cell and ECM). Variable $S_{\sigma_1 \sigma_2}$ is the area of the boundary between cells or between one cell and the ECM. Essentially, the model's first two terms tend to minimize the cell's area while keeping its volume constant shaping the cell into a sphere. Adhesion terms, however, try to maximize the boundary areas, flattening the cells. The competition between these terms makes individual cell look like a dome on the substrate (this is how we choose relative strengths of the cortex tension and characteristic adhesion), and two cells – like two domes pressed into each other side-by-side. To make the cells move, we must add the forces pushing the cell front and pulling its rear. Note that those forces originate from the cytoskeleton inside the cells, and not in the environment surrounding the cells, so the force balances are implied. Specifically, the force of protrusion that pushes on the cell leading surface forward from inside is balanced by a reactive cytoskeletal pushing directed to the rear and applied to the firm adhesions between the ventral surface of the cell and ECM. Similarly, the force of retraction that pulls the cell rear forward from inside is also balanced by a reactive cytoskeletal pulling directed to the rear and applied to the firm adhesions between the ventral surface of the cell and ECM.

We introduce these forces through effective potential energies as follows. First, we define a polarity for each cell, which is quantified using the angle between the polarization direction and the positive-x direction, $\varphi$, as shown in *Figure 4A*. Then, the respective potential energies can be defined as

$$W_p \left( \vec{r} \right) = -\Theta \left[ \alpha_p - |\alpha - \varphi| \right] |\vec{r} - \vec{r}_{COM}| Pro \left( \alpha - \varphi \right)$$

$$W_r\left(\vec{r}\right) = \Theta\left[|\alpha - \varphi| - \alpha_r\right]|\vec{r} - \vec{r}_{COM}|Ret$$

Here, $\Theta$ is the Heaviside step function (equal to 1/0 for positive/negative values of argument, respectively). $\vec{r} = (x, y, z)$ is the 3D position of a specific pixel. $\alpha$ is the angle of vector $(x - x_{COM}, y - y_{COM})$, in which $(x, y)$ is the position of 2D projection of a specific pixel onto the x-y-plane, and $(x_{COM}, y_{COM})$ is the 2D position of the centroid of the cell. $\varphi$ is the polarity angle mentioned above . $\alpha_p$ and $\alpha_r$ are the angular ranges of the protrusive and retractive forces, respectively. For example, if the protrusive force exists in the front half of the cell and the retractive force exists in the back half of the cell, then the angles are, $\alpha_p = \frac{\pi}{2}$ , $\alpha_r = \frac{3\pi}{2}$. $|\vec{r} - \vec{r}_{COM}|$ is the 3D distance between a specific pixel and the centroid of the cell and taking a gradient of it, results in the centripetal retractive and centrifugal protrusive forces. $Pro\left(\alpha - \varphi\right)$ and $Ret$ define the amplitudes of the energy terms that are also strengths of the protrusive and retractive forces, respectively. $Ret$ is a constant parameter, while $Pro$ is constant in some simulations but is a function of angle $(\alpha - \varphi)$ in others. Their values, as well as the values for $\alpha_p$ and $\alpha_r$ , and for all other model parameters, are listed in Appendix 1—table 1.

When investigating directionality and persistence of the cells' trajectories, stochasticity is introduced to the polarity's dynamics as follows:

$$d\varphi = -\omega_1\varphi\, dt - \omega_2\left(\varphi - \theta\right) + \sigma\, dW_t$$

where $\theta$ is the angle shown in **Figure 3B** and $dW_t$ denotes a Wiener process (stochastic directional noise). The first term shows the tendency of the polarity to align with the external signal's direction (the positive-x direction), and the second term shows the tendency to follow the other cell. For different polarization modes, $\omega_1$ and $\omega_2$ take different values. More specifically, for the independent mode and the faster-slower mode shown in **Figure 3B** (when the cells follow the environmental directional guidance independently), $\omega_1 \neq 0$, $\omega_2 = 0$ for both cells, while for the leader-trailer mode (when the trailing cell follows the leader instead of following the environmental guidance), $\omega_1 \neq 0$, $\omega_2 = 0$ for the leader and $\omega_1 = 0$, $\omega_2 \neq 0$ for the trailer.

It is worth mentioning that the exact absolute values of parameters in the energy function are not important as the dynamics of the system is determined by the ratio $\frac{\Delta H}{T}$ in which $T$ is a 'temperature' parameter without direct relation to the biological processes, and the 'Monte Carlo step' in the simulation is not directly related to an actual time scale. So, we only check whether the ratios of the model parameters are consistent with the experimentally estimated orders of magnitude of the biophysical parameters. Experimental estimates of the force generated over $1\mu m$ of the lamellipodial leading edge are $\sim 1000pN$ and the total traction force exerted by the cell is $\sim 10^4 - 10^5 pN$ (**Gline et al., 2015** ; **Mogilner and Oster, 2003**). As the leading edge of the lamellipodia is only $\sim 0.1 - 0.2\mu m$ thick, while in our model we cannot generate very thin protrusions, we distribute the total forces generated by the lamellipodia almost uniformly to the whole front of the cell and use the protrusive force density $\sim 100pN/\mu m^2$ assuming the height of the cell is $\sim 10\mu m$ (**Figure 1A**). Similarly, we distribute the total traction force uniformly to the back of the cell resulting in the retractive force density $\sim 10^2 - 10^3\, pN/\mu m^2$ . Thus, the orders of magnitude of the protrusive and retractive forces are close, and we keep them close in the model. The energy of adhesion between the cell and the substrate is estimated as follows. Each integrin attachment complex has a force $\sim 10 - 30pN$ (**Ananthakrishnan and Ehrlicher, 2007**) associated with it. The size of an integrin-based adhesion complexes formed at cell contacts with the ECM is $\sim 1\,\mu m^2$ (**Geiger et al., 2001**), so we estimate the adhesion force densities as $\sim 10 - 30\, pN/\mu m^2$ . Then, the ratio between the active forces and the adhesion forces is ~10. In our model, the force strength parameters, $Pro\left(\alpha - \varphi\right)$ and $Ret$, are on the order of 10–100 in dimensionless units, and the adhesion strength parameter $J$ ranges from 0 to 20 in dimensionless units, which is consistent with the force ratios from the experimental measurements.

After the orders of magnitude of the protrusion, retraction and adhesion energies are chosen as described, the rest of the principal model parameters are chosen following the following logic. The cortex contractility parameter $\kappa$ is chosen so that an individual non-motile cell has a shape close to that of a hemisphere; if this parameter is too small, the cell becomes a 'pancake'; if too large – a 'ball.' The parameter regulating the tightness of the cell volume control, $\lambda$ , is fine-tuned to avoid (1) freezing the cell shape – when this parameter has a value that is too great, most fluctuations of the cell shape get arrested, and (2) loosening the cell shape too much – when this parameter has a value that is too small, cell transiently becomes too small or too large that disagrees with the observations. The values

of the parameters for the stochastic directionality experiment are chosen so that the persistence of the single cell predicted trajectories fit that of the observed trajectories. Finally, note that the parameters for the adhesion strength (J) are scaled as follows. We make this parameter a large positive number for the boundary between the two motile cells and endoderm (or for cell-free space boundary in simulations without endoderm); this corresponds to the 'no adhesion' regime. Then, the adhesion parameters for cell-cell and cell-ECM boundaries are smaller positive numbers. Thus, the energy in the system decreases when the relative areas of the cell-cell and cell-ECM boundaries increases, so those are the adhesive surfaces.

Note that the dynamics of the simulated cell is determined by the probability function of spin changes, which is defined by the exponential function with a cutoff at 1. This leads to a speed-force relation of an exponential form when the force is too small and is saturated when the force is too large. We avoid this artifact because the parameters we choose restrict our simulations to the regime where the speed-force relation is approximately linear.

Simulations were done using software CompuCell3D 3.7.8 (*Swat, 2012*). In Appendix 1 tables, we list all model parameters that are varied between different simulations, and in the Supporting information we explain the reasons for the variance. When we simulate the actively migrating cells in the presence of the endoderm that mechanically resists the deformations, we make the endodermal cells mechanically more passive than the migrating cells (their contractile tension is half that of the migrating cells) and vary the endodermal 'tightness of the volume conservation' parameter $\lambda$ a few fold less and greater than that of the migrating cells, respectively.

## Acknowledgements

We thank Alex McDougall for his generous gift of the iMyo reporter. This work was supported by NIH F32 GM108369-01A1 postdoctoral fellowship to YYB, NIH/NIGMS GM096032-09 award to LC, AM, and HY. US Army Research Office grant W911NF-17-1-0417 to AM and HY, NSF grant DMS-1950981 to CC and AM.

## Additional information

### Funding

| Funder | Grant reference number | Author |
| --- | --- | --- |
| National Institute of General Medical Sciences | GM108369-01A1 | Yelena Y Bernadskaya |
| National Institute of General Medical Sciences | GM096032-09 | Lionel Christiaen |
| Division of Mathematical Sciences | DMS-1950981 | Alex Mogilner |
| U.S. Army | W911NF-17-1-041 | Alex Mogilner |

The funders had no role in study design, data collection and interpretation, or the decision to submit the work for publication.

### Author contributions

Yelena Y Bernadskaya, Conceptualization, Formal analysis, Funding acquisition, Investigation, Methodology, Validation, Visualization, Writing – original draft, Writing – review and editing; Haicen Yue, Conceptualization, Formal analysis, Investigation, Methodology, Software, Validation, Writing – original draft; Calina Copos, Conceptualization, Visualization; Lionel Christiaen, Conceptualization, Funding acquisition, Supervision, Writing – original draft, Writing – review and editing; Alex Mogilner, Conceptualization, Funding acquisition, Resources, Supervision, Writing – original draft, Writing – review and editing

### Author ORCIDs

Yelena Y Bernadskaya http://orcid.org/0000-0001-6147-5825
Lionel Christiaen http://orcid.org/0000-0001-5930-5667

Alex Mogilner http://orcid.org/0000-0002-9310-3812

**Decision letter and Author response**
Decision letter https://doi.org/10.7554/eLife.70977.sa1
Author response https://doi.org/10.7554/eLife.70977.sa2

## Additional files

### Supplementary files
• Transparent reporting form

### Data availability
The code generated during this study is available on GitHub at https://github.com/HaicenYue/3D-simulation-of-TVCs, copy archived at https://archive.softwareheritage.org/swh:1:rev:c8a79bc7822f295ab72edb8e3f7660c823c3699e).

The following dataset was generated:

| Author(s) | Year | Dataset title | Dataset URL | Database and Identifier |
|---|---|---|---|---|
| Yue H | 2021 | 3D-simulation-of-TVCs | https://github.com/HaicenYue/3D-simulation-of-TVCs.git | Github, Github |

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

# Appendix 1

**Appendix 1—table 1.** Parameterization of cell-shaping forces, cell adhesion, polarization, and endoderm stiffness for single migrating cells and cell pairs.

| Parameter name | Standard single | Supracellular double | Double same | Climbing |
|---|---|---|---|---|
| Used in figure | *Figure 1B and C* | *Figures 1B, C and 2B* (after polarization), *Figure 3D* (LT mode), *Figure 3G* | *Figures 1E and 2B* (before polarization), *Figure 3D* (I mode, FS mode) | *Figure 2—figure supplement 1* |
| $\lambda$ | 0.1 | 0.1 | 0.1 | 0.2 |
| $V$ | 905 | 905 | 905 | 905 |
| $\kappa$ | 0.02 | 0.02 | 0.02 | 0.06 |
| $J_{LT}$ | | 16 | 16 | 16 |
| $J_{LS}$ | 14 | 14 | 14 | 14 |
| $J_{TS}$ | | 15 | 14 | 40 |
| $Pro_L$ | 160 | 180 | 160 (200 for FS mode in *Figure 3D*) | 180 |
| $Ret_L$ | 40 | 20 | 40 (50 for FS mode in *Figure 3D*) | 20 |
| $Pro_T$ | | $150 \left( \alpha - \varphi \right)$ * | 160 | $150 \left( \alpha - \varphi \right)$ * |
| $Ret_T$ | | 70 | 40 (30 for FS mode in *Figure 3D*) | 300 |
| $\alpha_{pL}$ | 66° | 90° | 66° | 90° |
| $\alpha_{rL}$ | 90° | 90° | 90° | 90° |
| $\alpha_{pT}$ | | 90° | 66° | 90° |
| $\alpha_{rT}$ | | 90° | 90° | 90° |
| **With noise** | | | | |
| Used in figure | *Figure 4A and B* | *Figure 4A and B* | | |
| $\omega_1$ | 0.005 | 0.005 | | |
| $\omega_2$ | | 0.1 | | |
| $\sigma$ | 0.1 | 0.1 | | |
| **With endoderm cells** | | | | |
| Used in figure | *Figure 5D* | | *Figure 5D* | | *Figure 5D* | |

| | Soft | Stiff | Soft | Stiff | Soft | Stiff |
|---|---|---|---|---|---|---|
| $\lambda_E$ | 0.05 | 0.5 | 0.05 | 0.5 | 0.05 | 0.5 |
| $V_E$ | 1,000 | 1,000 | 1,000 | 1,000 | 1,000 | 1,000 |
| $\kappa_E$ | 0.01 | 0.01 | 0.01 | 0.01 | 0.01 | 0.01 |
| $J_{EE}$ | 0 | 0 | 0 | 0 | 0 | 0 |
| $J_{LT}$ | | | 10 | 10 | 16 | 16 |

All the other $J$ s not listed above are 20.
The $J$ s not listed for the 'with noise' and 'with endoderm' part are the same as the part without noise or endoderm.
$L, T, E, S$ in the subscripts of parameter names mean 'leader,' 'trailer,' 'endoderm,' and 'substrate,' respectively.
* $\left( \alpha - \varphi \right)$ is as listed in equations in the Materials and methods section.

**Appendix 1—table 2.** Parameterization of migration mode of three migrating cells.

| Parameter name | Three cells: independent mode and leader-mid-trailer mode | Three cells: faster-slower mode |
|---|---|---|
| Used in figure | *Figure 3E* | *Figure 3E* |
| $\lambda$ | 0.1 | 0.1 |
| $V$ | 905 | 905 |
| $\kappa$ | 0.02 | 0.02 |
| $J_{LM}$, $J_{LT}$, $J_{MT}$ | 16 | 16 |
| $J_{LS}$, $J_{MS}$, $J_{TS}$ | 15 | 15 |
| $Pro_L$ | 160 | 170 |
| $Ret_L$ | 40 | 45 |
| $Pro_M$ | 160 | 150 |
| $Ret_M$ | 40 | 40 |
| $Pro_T$ | 160 | 120 |
| $Ret_T$ | 40 | 30 |
| $\alpha_{pL}$ | 66° | 66° |
| $\alpha_{rL}$ | 90° | 90° |
| $\alpha_{pM}$ | 66° | 66° |
| $\alpha_{rM}$ | 90° | 90° |
| $\alpha_{pT}$ | 66° | 66° |
| $\alpha_{rT}$ | 90° | 90° |

All the other $J$ s not listed above are 20.
$L$, $T$, $M$, $S$ in the subscripts of parameter names mean 'leader,' 'trailer,' 'middle,' and 'substrate,' respectively.

## Arguments for choosing specific spatial-angular force distributions
### Single cell
Before we determine the force distribution for the TVC pair, we first study the force distribution in a single cell. As shown in *Figure 1—figure supplement 1A*, if the retractive force is concentrated narrowly at the very rear of the cell (if viewed from above), then the cell shape deviates from that observed. Similarly, if the protrusive force is concentrated in the narrow layer near the flat substrate, the front of the cell deforms into unstable wavy shapes not resembling the observations (*Figure 1—figure supplement 1B*). Therefore, we chose to distribute the retractive force spatially over the whole rear half of the cell. Two simplest choices for orientation of the retractive forces are (1) parallel to the substrate and (2) centripetal (the force is inward, along the radial lines from the cell centroid). We decided in favor of the second option because (1) this option recapitulates the observed tapered cell rear with better than the parallel-to-the-surface force, and (2) the origin of this force is likely stress fibers connecting the rear dorsal and central ventral surfaces of the cell, and this geometry is closer to the second option.

   One attractive hypothesis would be that the protrusive force is generated by actin filament polymerization of the thin lamellipodial network at the ventral surface of the cell. However, if we restrict the protrusive force only near the bottom of the cell, we must increase the magnitude of the protrusive force by a few fold in order to maintain its total contribution. After a simulation with such protrusive restricted to the bottom one or two pixels, with a magnitude equaling 400, the shape of the cell becomes wavy and unstable, as shown in *Figure 1—figure supplement 1B*. Thus, we relax this restriction and assume that the protrusive force is oriented centrifugally. This choice, of course, raises the question of how such forces are generated. One possibility is that

the origin of this force is an isotropic swelling of the gel (cytoskeleton). Future experiments will have to address this question. Then, we need to determine the distribution of the forces in the x–y plane. In principle, the protrusive forces in the x–y plane can be parallel to the x-axis. However, if the magnitude of such force does not depend on the y coordinate, we observed in the simulation that the cell leading edge and side become flat and too wide. Therefore, to avoid introducing an additional parameter to grade the force, we chose the radial centrifugal distribution of the protrusive force in the x–y plane, which reproduces the characteristic leading edge shape without an additional parameter.

In *Figure 1—figure supplement 1A*, we show the typical cell shapes for protrusive and retractive forces distributed within different ranges. We find that decreasing the range of retractive force or increasing the range of protrusive force makes the cell wider and oppositely, the cell becomes longer. Based on the aspect ratio, we finally decide that the retractive force is distributed uniformly in x–y plane in the whole back half of the cell while the protrusive force is distributed uniformly in x–y plane within the range of angle $\alpha = 66°$ as shown in *Figure 1D*.

## TVC pair

TVC pair is not a simple combination of two independent cells. We have shown in *Figure 1E* that for two independent same cells moving together the leading edge of the trailer cell is too wide, which suggests the protrusive force is too widely distributed, and the polarity of the cell pair is not well maintained. This implies that when two cells move together, their force distribution changes because of the cell-cell interactions. So, we decrease the protrusive force in the trailer cell and make it more concentrated along the central axis of the cell by multiplying the magnitude of the force by $\alpha$ . This is also consistent with the observation in vivo that in most cases the two cells' junction is concave if we look at it from the trailer cell. In addition, we need to increase the retractive force in the trailer cell accordingly to maintain the total force approximately the same so that the speed of the trailer cell does not decrease a lot. This is also reasonable biologically as the generation of protrusive and retractive force shares some same cytoskeletal components. We also show in *Figure 1—figure supplement 1C* that if we do not increase the retractive force accordingly, the trailer cell moves slower and will detach from the leader cell. (We observed in general that when the sum of the protrusive and retractive force in the trailer deviates too much from that in the leader, the cells split apart.) We can also observe in vivo that the shape of the leader cell is quite different from a single cell (*Figure 1B*). The TVC pair as a whole has a similar aspect ratio as a single cell (*Figure 1C*) while the leader cell itself becomes wide and short. This implies that, contrary to the trailer cell, in the leader cell the contribution of the protrusive force increases while the contribution of the retractive force decreases. So, we decrease the magnitude of the retractive force in the leader cell and increase that of the protrusive force while at the same time make it more widely distributed in the cell by setting $\alpha = 90°$. We also show in *Figure 1— figure supplement 1D* that if we keep the force distribution in the leader cell the same as that in the single cell, the leader cell is too long and thus the cell pair looks more like two independent cells connected rather than a supracellular pair, with the aspect ratio similar to a single cell. This is the train of arguments for determining the force distribution in a cell pair based on that in a single cell.

Our main idea is that when two cells are moving as a cell pair, they redistribute their forces by decreasing the force near the cell-cell junction while increasing the forces at the front and back of the pair. But the forces near the junction cannot be decreased too much. We show in *Figure 1— figure supplement 1E* that if the retractive force in the leader and the protrusive force in the trailer are decreased to zero, we will have a leader cell with a long tail as there is not enough force to push the cell-cell junction forward.

It is worth mentioning that the model predicts qualitatively reasonable shapes and movements not only for the chosen set of parameters; varying most of the parameters a few fold still gives reasonable results. Variation of the model parameters from one numerical experiment to another is shown in Appendix 1 tables; mostly, the logic of such variation should be clear from the description of the numerical experiment. Just two notes about the cases when this logic may not be so transparent: when we simulate the trailer 'climbing' the 'leader,' not only the adhesion of the trailer to the substrate is decreased, but also the tightness of the volume conservation and the surface tension are increased, otherwise the cell at the bottom becomes too flat. When we simulate cells migrating under the endoderm, the cell-cell adhesion is adjusted so that the characteristic supracellular shape is conserved.

## Varying endoderm stiffness

In the model, two parameters, $\lambda$ and κ, determine cell mechanical properties. Parameter $\lambda$ characterizes how tight cell volume control is, while parameter κ characterizes the tension of the cell cortex. We define the mechanical properties of the endodermal cells relative to those of the TVCs. Specifically, we set parameter κ for the endodermal cells twice smaller than that for the TVCs assuming that the quiescent endodermal cells are not as contractile as the TVCs. We call the endoderm 'soft' if its cells' parameter $\lambda$ is twice smaller than that for the TVCs – in this case, the endodermal cells are more readily deformed and pushed out of the way by the migrating cells. We call the endoderm 'stiff' if its cells' parameter $\lambda$ is fivefold greater than that for the TVCs – in this case, the endodermal cells resist the deformations. Thus, the stiff cells have one order of magnitude tighter volume control than the soft cells and migrating TVCs in the model have deformation properties between the soft and stiff endodermal cells.

