## [Decision Letter]

**Decision letter after peer review:**

Thank you for submitting your article "Supracellular organization confers directionality and mechanical potency to migrating pairs of cardiopharyngeal progenitor cells" for consideration by *eLife*. Your article has been reviewed by 2 peer reviewers, and the evaluation has been overseen by a Reviewing Editor and Naama Barkai as the Senior Editor. The following individuals involved in review of your submission have agreed to reveal their identity: Ajay Chitnis (Reviewer #2).

As you will see, the reviewers found your paper interesting but raised some concerns. Please address them in full before resubmitting.

*Reviewer #1:*

In this manuscript, authors develop a computational model of migrating pairs of cardiopharyngeal progenitors in Ciona model to describe their properties during migration. Most of the predictions drawn from the simulations are corroborated experimentally in vivo. The polarized migrating pair of progenitors represent a minimum unit of collective migration showing essentially that "two cells are better than one".

The simulation of the migrating cell pair is based on Cellular Potts Model and uses cell morphology as a proxy to model different mechanical and cellular forces that generate specific cell shapes during migration. The model seems to faithfully reproduce the in vivo observations. The arguments describing the model are well explained. The model thus provides a number of predictions pertaining the migrating cell collectives that can be tested in vivo using genetic and molecular tools. The authors then test some of the predictions prompting them to conclude that the migrating progenitor pair present the simplest model of collective migration, in which the two cells are intercellularly coupled and their polarization occurs across the two-cell continuum, thus forming a supracellular collective. This cell collective displays hierarchy, which favors linear arrangement conferring directionality and persistence in migratory behavior. While some conclusions are well justified and the experimental evidence corroborates the simulations, at few points the conclusions are more speculative; the figure panels not always match the manuscript text. The concept that the migrating pair can deform the overlaying endoderm is the least developed and not probed in vivo. In summary, the computational model represents a powerful tool to examine directional migration of polarized cell collectives with similar characteristics allowing for studying how biomechanical cues integrate with molecular signals across group of cell collectives to coordinate their behavior.

In Figure1: The manuscript text does not follow the arrangements of figure panels. First data corresponding to 1A, C, D are described and only then to 1B. Then 1F is described before 1E. Also in the Figure 1F is mislabeled as 1G. This is all quite confusing to follow.

The shape measurements should be described in a schematic. The L/T sphericity label in y-axis of graph in C suggests it is a ratio, but the overall shape was measured. In D, the bounding box with the length and width should be indicated. Were all measurements performed on dorsal views?

In Figure 1E, the non-muscle myosin distribution has to be better quantified. While the line plots suggest the depletion from the leader-trailer cell membrane junction, myosin appears to be accumulating at the front of cell periphery in both leader and trailer. The claim that it is relatively depleted is not entirely true when looking at the example presented. What is the myosin distribution in the sagittal sections? What is the distribution of actin? The more detailed actomyosin distribution would further support the model in 1B. On what data are the conclusions about absence of "contractility organized in supracellular fashion" based on? It seems that leader overall may have less non-muscle myosin then trailer, suggesting that maybe there is a polarized distribution of myosin.

In Figure 2: In the first paragraph, the concept of supracellular polarity between leader and trailer should be introduced as a hypothesis. At the current state it seems that just the change in their shape justifies the claim that the pair is supracellularly polarized.

In 2A, graph at right, when is the sphericity measured, only when DdrDN is expressed in trailer or leader? This needs to be clarified.

Because the authors in these sets of experiment also point out that increased protrusive behavior is more likely adopted in the leader cell, the conclusions drawn here in this section about the polarization at the supracellular level have to be taken this data into account. In similar way as presented in the introduction to Figure 3.

Figure 3:

Similar to Figure 1, the figure panels do not completely follow the manuscript text. Please revise.

The fact that the simulation predicts the hierarchy in the linear arrangements is one of the key findings. The presentation of the in vivo data in Figure 3E are however not so easy to follow. The confusing part is to follow the numbers (in %) between the manuscript text and the graphs in 3E. If the alignment occurs only at 57%, isn't it then random?

What is the unit of displacement of y-axis in 3H bottom graph.

Figure 5:

The concept of endoderm deformation has to be clarified. The readers that are not directly from the field of TCV migration may not necessarily understand it. Do the authors mean the little pocket that endoderm forms around TCVs shown in merge of 5A? If that is the case, please add arrowheads or similar to highlight this better. The deformation is then happening locally at the cellular level. The term “deforming endoderm” implies the deformation at the tissue-scale. This needs clarification.

What is the evidence for differing stiffness of endoderm and epidermis?

What are the values and units for “varying endoderm stiffness”? At the minimum some relative value should be added here to understand the difference in the odelling parameters. Are the assumptions about the stiffness based on physiological values?

If the endoderm is soft, to what extent it is important to consider its mechanical resistance? Afterall, the 2-cell equivalent significantly outperformed the supracellular arrangements with soft endoderm. This should be commented.

While interesting to use the modelling to understand the forces from the surrounding tissue exerted on the migrating cell pair, the concept here is the least developed and not tested experimentally. The sentence in the abstract referring to this data is somewhat misleading and needs revising.

Statistics: in some experiments the N number of biological replicates is indicated, but not in all experiments. This should be corrected. In the methods section, Mann-Whitney test is described as used to compare two experimental groups, but in some panels t-test with Welch’s correction is used. It should be clarified and corrected. What software/tool was used to perform statistical tests? The in vivo data should be presented with S.D. not S.E.M.

Overall, it is very interesting study, and the power of the computational simulation is very well reflected in the in vivo experiments. My main recommendation for improving the manuscript (in addition to above points) is to re-write the flow, to simplify the motivations for the experiments, and conclusions drawn from it, and leave the speculations and assumptions for the discussion.

*Reviewer #2:*

In this study, the authors develop a model to understand the collective migration of migrating pairs of cardiopharyngeal progenitor cells. This represents an simplest form of collective migration with two cells. They propose that the collective migrates as a “supracell”, with leader cells assuming a greater protrusive capability and trailer cells assuming greater retractive capability. They meticulously study the effects of leader-trailer and cell-matrix adhesivity, intracellular force distributions and noise on robustness of cell migration. They corroborate their simulation results with experiments. Overall, this study comprehensively demonstrates that migrating as a collective leads to more mechanically efficient and persistent migration than as a single motile cell. A particular strength of the paper is that the authors have done an excellent job of explaining how the Cellular Potts Model works and how they chose to represent specific biological details using this modelling environment. The model is also likely to provide a useful framework for development of models of more complex examples of collective migration.

Specific comments questions:

1. Page 3, para 1 – "TVCs maintain polarized and bulky shapes as they invade…". It would be helpful if the authors could clarify what they mean by polarized and bulky.

2. Quantification and Statistical Analysis, Page 14 – What was the size of Gaussian filter chosen by the authors to preprocess their images?

3. How would the authors biophysically justify their choice of protrusive force strength being a function of (α – φ), while retractive force strength is a constant value? Would simulation results differ significantly if retractive force strength in trailer or single cells is a function of (β – φ) or (90⁰ – φ)?

---

## [Author Response]

Reviewer #1:In this manuscript, authors develop a computational model of migrating pairs of cardiopharyngeal progenitors in Ciona model to describe their properties during migration. Most of the predictions drawn from the simulations are corroborated experimentally in vivo. The polarized migrating pair of progenitors represent a minimum unit of collective migration showing essentially that "two cells are better than one".The simulation of the migrating cell pair is based on Cellular Potts Model and uses cell morphology as a proxy to model different mechanical and cellular forces that generate specific cell shapes during migration. The model seems to faithfully reproduce the in vivo observations. The arguments describing the model are well explained. The model thus provides a number of predictions pertaining the migrating cell collectives that can be tested in vivo using genetic and molecular tools. The authors then test some of the predictions prompting them to conclude that the migrating progenitor pair present the simplest model of collective migration, in which the two cells are intercellularly coupled and their polarization occurs across the two-cell continuum, thus forming a supracellular collective. This cell collective displays hierarchy, which favors linear arrangement conferring directionality and persistence in migratory behavior. While some conclusions are well justified and the experimental evidence corroborates the simulations, at few points the conclusions are more speculative; the figure panels not always match the manuscript text. The concept that the migrating pair can deform the overlaying endoderm is the least developed and not probed in vivo. In summary, the computational model represents a powerful tool to examine directional migration of polarized cell collectives with similar characteristics allowing for studying how biomechanical cues integrate with molecular signals across group of cell collectives to coordinate their behavior.

We would like to thank the reviewer for their positive evaluation of our work. Detailed response follows below.

In Figure1: The manuscript text does not follow the arrangements of figure panels. First data corresponding to 1A, C, D are described and only then to 1B. Then 1F is described before 1E. Also in the Figure 1F is mislabeled as 1G. This is all quite confusing to follow.

The figure has been updated to flow in the order of presentation.

The shape measurements should be described in a schematic. The L/T sphericity label in y-axis of graph in C suggests it is a ratio, but the overall shape was measured.

We have added a supplemental figure that contains a schematic of sphericity calculations as well as a supplemental table of in vivo measurements of leader/trailer sphericities.

In D, the bounding box with the length and width should be indicated. Were all measurements performed on dorsal views?

All measurements are performed on dorsally oriented views of the cells at embryonic stage 23. This has now been included in the methods section and height and width of bounding boxes, normalized to the width, have been added to the figure now labeled 1C.

In Figure 1E, the non-muscle myosin distribution has to be better quantified. While the line plots suggest the depletion from the leader-trailer cell membrane junction, myosin appears to be accumulating at the front of cell periphery in both leader and trailer. The claim that it is relatively depleted is not entirely true when looking at the example presented.

Thank you for your comment. We have further quantified the enrichment of Myosin at the cell-cell junction and find that it does not differ between cytoplasmic regions of the leader and trailer cells and the area of the cell-cell junction, as assayed by line scans across the junction taking the average intensity readings of the membrane marker and the myosin reporter. While membrane markers naturally accumulate at the cell-cell junction resulting in higher average fluorescence intensity at the junction, no myosin accumulation was detected at the junction or in adjacent cytoplasmic regions. This data is now included in Supplemental Figure 4.

What is the myosin distribution in the sagittal sections?

We have added Supplemental Figure 4 that includes a representative sagittal image of myosin distribution.

What is the distribution of actin?

We have added Figure 2D, a section that shows analysis of Lifeact::GFP distribution in leader/trailer cell pairs. Here we find that Lifeact puncta, generally understood to represent adhesion complexes that are connected to filamentous actin, are enriched in the leader compared to the trailer. This additional information fits well with the prediction of the model and our in vivo analysis of reduction of adhesion in one of the cells in the pair, suggests that higher adhesion is indeed a property of the leading cell. The additional paragraph is highlighted in the text as well as the description of analysis in the methods section.

The more detailed actomyosin distribution would further support the model in 1B.

We have now added additional information about myosin distribution as well as distribution of ventral actin enrichment (Supplemental Figure 4 and Figure 2D). While we are not able to describe protrusion dynamics in this manuscript, adding information regarding unequal distribution of actin puncta to the leading cell does support our model, particularly the higher adhesive activity of leading cells compared to the trailers.

On what data are the conclusions about absence of "contractility organized in supracellular fashion" based on? It seems that leader overall may have less non-muscle myosin then trailer, suggesting that maybe there is a polarized distribution of myosin.

Thank you for your comment. We clarified the language which now reads as follows: However, in contrast to neural crest cells and *Drosophila* border cells, we did not observe supracellular actomyosin “cables” extending across cardiopharyngeal progenitors and there is no exchange of cell positions between TVCs.

In Figure 2: In the first paragraph, the concept of supracellular polarity between leader and trailer should be introduced as a hypothesis. At the current state it seems that just the change in their shape justifies the claim that the pair is supracellularly polarized.

Thank you for your comment. The language has been clarified.

In 2A, graph at right, when is the sphericity measured, only when DdrDN is expressed in trailer or leader? This needs to be clarified.

We did not include a marker for following the inheritance of Ddr^dn^ in this experiment. Our empirical knowledge of the system suggests that under the amount of plasmid used in this experiment approximately 60% TVC pairs will inherit the perturbation inducing plasmid in both leader **and** trailer cell. 40% will inherit the plasmid in either the leader **or** the trailer, meaning that for any given cell for which sphericity is measured, under these conditions there is an 80% chance that it has inherited the Ddr^dn^-expressing plasmid. Overall, this does not reduce the sensitivity of our assays. Clarifying statement has been added to methods and has been highlighted and percent incidence is now shown in figure 2C. It now reads as follows: For experiments described in Figure 2A no marker was used to follow transmission of Foxf>Ddr^nd^ and all cells were analyzed. Under these conditions there is an 80% chance that any given cell has inherited the Ddr^dn^ perturbation.

Because the authors in these sets of experiment also point out that increased protrusive behavior is more likely adopted in the leader cell, the conclusions drawn here in this section about the polarization at the supracellular level have to be taken this data into account. In similar way as presented in the introduction to Figure 3.

Thank you for your comment. We have included two sentences in the introduction to figure 2 integrating the generation of the leading edge and protrusive activity and development of TVC leader/trailer polarization. The change is highlighted in red and now reads as follows: Cell-matrix interactions and distribution of protrusive activity to the leading TVC result in the generation of a broad leading edge, during the establishment of collective TVC polarity 31. The generation of flattened edge lowering leader cell sphericity. Conditions that perturb cell-matrix adhesion often increase the sphericity of leading cells, suggesting they fail to establish flattened protrusions at the leading edge31.

Figure 3:Similar to Figure 1, the figure panels do not completely follow the manuscript text. Please revise.

Thank you for your comment. We have edited the figure to flow in the order of discussion in the text.

The fact that the simulation predicts the hierarchy in the linear arrangements is one of the key findings. The presentation of the in vivo data in Figure 3E are however not so easy to follow. The confusing part is to follow the numbers (in %) between the manuscript text and the graphs in 3E. If the alignment occurs only at 57%, isn't it then random?

Thank you for the opportunity to clarify our findings. When the B7.5 cells are all converted to the migratory TVC fate the cells are initially arranged in a square, as shown in Figure 3G, bottom micrograph. When these cells begin to migrate, the possible number of permutations of their arrangement is much larger than just aligned or unaligned. As we can see from the simulations, specific conditions limiting and constraining the movement of cells must exist to generate the linear alignment in a reasonable amount of time that it takes the cells to migrate in vivo. We therefore conclude that the emergence of linear arrangements is not a random event.

What is the unit of displacement of y-axis in 3H bottom graph.

The unit is equal to the size of the computational pixel in the model. The approximate volume of the cell is 10x10x10 pixels, so considering that the approximate cell dimension is ~ 10 micron, the unit is roughly 1 micron. We added the notation to figure now labeled 3G.

Figure 5:The concept of endoderm deformation has to be clarified. The readers that are not directly from the field of TCV migration may not necessarily understand it. Do the authors mean the little pocket that endoderm forms around TCVs shown in merge of 5A? If that is the case, please add arrowheads or similar to highlight this better.

Thank you for your comment. We’ve added arrows to the micrograph to draw attention to the endodermal depression pocket left by the TVCs during migration.

The deformation is then happening locally at the cellular level. The term "deforming endoderm" implies the deformation at the tissue-scale. This needs clarification.

Thank you for your comment. While the endoderm does consist of individual cells, the TVCs displace multiple endodermal cells at the same time, therefore affecting more than a single cell. While this undoubtedly exerts some compression force on the endoderm there is indeed no indication that it propagates to the rest of the large endodermal tissue. We have clarified the language to stress that the deformation is local.

What is the evidence for differing stiffness of endoderm and epidermis?

It is indeed true that no current data exists about the difference in stiffness between the endoderm and the epidermis. We base our assumptions on the following observations:

1. The epidermis does not become deformed by the migrating TVCs in any appreciable way. In contrast to this, the endoderm is displaced by the moving cells resulting in readily observable “pocket” depressions in the tissue, as shown in this paper, in Bernadskaya et al., 2019, and Gline et al., 2015.

2. In Bernadskaya et al., 2019 we report that the TVCs migrate over the collagen-containing extracellular matrix, not visualized in this paper. The TVCs only contact the ECM on their ventral surface, leaving the dorsal surface free of ECM or any adhesive structures we’ve visualized so far.

3. Limited information exists regarding the relative stiffness of embryonic tissues in Ciona. A recent study looking at early development of the Ciona embryo (112 cell stage) suggests that the cells committed to the epidermal fate at this stage are stiffer than the surrounding embryonic cells, although endodermal cells are also relatively stiff at this point due to gastrulation of the embryo which is undergoing invagination driven by apical constriction of the endodermal cells on the vegetal hemisphere of the embryo (Fujii, et al., 2021).

4. Multiple papers have shown that the mesenchymal cell migration, which is the type carried out by the TVCs, proceeds more efficiently on a stiff surface compared to a softer surface, with migratory cells preferentially staying on stiffer gel substrates where they can form stronger focal adhesions. (Raab, et al., 2012, Reviewed in Shellard and Mayor 2021). We believe this to be the case in Ciona embryos as well, although existence of mechanical gradients has not been experimentally demonstrated in vivo.

We therefore believe that the endoderm indeed represents a softer surface when compared to the epidermis. Direct measurements of tissue stiffness would be optimal but are unfortunately beyond the scope of this study.

What are the values and units for "varying endoderm stiffness”? At the minimum some relative value should be added here to understand the difference in the modeling parameters.

The mechanical parameters of the TVC cells and endoderm cells are listed in the Supplemental Table. We added the explanation for varying endoderm stiffness to the Supplemental Material and referred to it in the paper. In the model, two parameters, λ and κ determine cell mechanical properties. Parameter λ characterizes how tight is cell volume control, while parameter κ characterizes the tension of the cell cortex. We define the mechanical properties of the endodermal cells relative to those of the TVC cells. Specifically, we make parameter κ for the endodermal cells twice smaller than that for the TVC cells assuming that the quiescent endodermal cells are not as contractile as the TVC cells. We call the endoderm ‘soft’ if its cells’ parameter λ is twice smaller than that for the TVC cells – in this case the endodermal cells are more readily deformed and pushed out of the way by the migrating cells. We call the endoderm ‘stiff’ if its cells’ parameter λ is 5-fold greater than that for the TVC cells – in this case the endodermal cells resist the deformations. Thus, the stiff cells have one order of magnitude tighter volume control than the soft cells and migrating TVC cells in the model have deformation properties between the soft and stiff endodermal cells.

Are the assumptions about the stiffness based on physiological values?

This is primarily inferred from our observations that the epidermis does not deform upon passage of the TVCs in control embryos. There are no mechanical measurements available from Ciona embryo, and this is an important avenue for future work.

If the endoderm is soft, to what extent it is important to consider its mechanical resistance? Afterall, the 2-cell equivalent significantly outperformed the supracellular arrangements with soft endoderm. This should be commented.

If the endoderm is ‘soft’ (both parameters λ and kappa are smaller than those of the TVC cells), its mechanical resistance is negligible – in this case simulations without the endoderm give almost the same speeds as those with soft endoderm. Indeed, with soft endoderm, the 2 equivalent cells in the leader-trailer configuration slightly (we would not say significantly) outperform the supracellular cell pair. This changes significantly under the stiff endoderm, where presumably the supracellular pair has a more optimal shape (like the smoothened surface of a sports car, in a way). We added a comment to the text.

While interesting to use the modelling to understand the forces from the surrounding tissue exerted on the migrating cell pair, the concept here is the least developed and not tested experimentally. The sentence in the abstract referring to this data is somewhat misleading and needs revising.

We revised respective sentence in the abstract as follows:

“Simulations suggest that cell pairs can overcome mechanical resistance of the trunk endoderm more effectively when they are polarized collectively.”

Here we only refer to the simulations showing that the speed of the cell pair under the stiff endoderm is maximal when they are in the Leader-Trailer configuration, with protrusion-retraction forces organized in supracellular fashion.

Statistics: in some experiments the N number of biological replicates is indicated, but not in all experiments. This should be corrected.

Thank you for your comment. We have updated the figure legends to include the number of biological replicates for in vivo data.

In the methods section, Mann-Whitney test is described as used to compare two experimental groups, but in some panels t-test with Welch’s correction is used. It should be clarified and corrected.

Thank you for your comment. We have updated the Methods section to include a statement about simulation data analysis.

What software/tool was used to perform statistical tests? The in vivo data should be presented with S.D. not S.E.M.

Thank you for your comment. Statistics is indeed often misused in representing biological data, and we are happy to address this question. For all in vivo data GraphPad Prism software was used to generate graphs and perform statistical analysis. While we do show graphs with S.E.M, we have purposefully chosen to use scatter plots from which variability of each dataset can be estimated visually. Since our goal is to compare means between different conditions, not describe the variability of the system, S.E.M is a reasonable choice for visual representation, although we note that 95% C.I. is also a good option. This is outlined in the statistics guide of the GraphPad Prism software, which can be viewed here:

https://www.graphpad.com/guides/prism/latest/statistics/statwhentoplotsdvssem.htm

Statistical analysis in the manuscript is not affected by changing the display of descriptive statistics.

Overall, it is very interesting study, and the power of the computational simulation is very well reflected in the in vivo experiments. My main recommendation for improving the manuscript (in addition to above points) is to re-write the flow, to simplify the motivations for the experiments, and conclusions drawn from it, and leave the speculations and assumptions for the discussion.

Thank you for your comments. We firmly believe that most manuscripts benefit from thoughtful reviews and have now streamlined the text with your suggestions in mind. We have clarified the language in section introductions and removed speculative statements from the Results section.

Reviewer #2:In this study, the authors develop a model to understand the collective migration of migrating pairs of cardiopharyngeal progenitor cells. This represents an simplest form of collective migration with two cells. They propose that the collective migrates as a "supracell", with leader cells assuming a greater protrusive capability and trailer cells assuming greater retractive capability. They meticulously study the effects of leader-trailer and cell-matrix adhesivity, intracellular force distributions and noise on robustness of cell migration. They corroborate their simulation results with experiments. Overall, this study comprehensively demonstrates that migrating as a collective leads to more mechanically efficient and persistent migration than as a single motile cell. A particular strength of the paper is that the authors have done an excellent job of explaining how the Cellular Potts Model works and how they chose to represent specific biological details using this modelling environment. The model is also likely to provide a useful framework for development of models of more complex examples of collective migration.

We are grateful to the reviewer for their positive evaluation of our work. Detailed responses are below.

Specific comments questions:1. Page 3, para 1 – "TVCs maintain polarized and bulky shapes as they invade…". It would be helpful if the authors could clarify what they mean by polarized and bulky.

The text has been clarified and now reads as follows: TVCs maintain polarized organization along the anterior posterior axis and spheroid shapes as they invade the extracellular space between the epidermis and the endoderm.

2. Quantification and Statistical Analysis, Page 14 – What was the size of Gaussian filter chosen by the authors to preprocess their images?

Size of the Gaussian filter applied was between 0.06 and 0.08 based on the background variability of the individual images. This information has been added to Materials and methods.

3. How would the authors biophysically justify their choice of protrusive force strength being a function of (α – φ), while retractive force strength is a constant value? Would simulation results differ significantly if retractive force strength in trailer or single cells is a function of (β – φ) or (90⁰ – φ)?

We use the constant, angle-independent protrusion and retraction forces in most cases. The only case in which the protrusion force becomes angle-dependent is *at the front of the trailer cell*, when the cells move in the supracellular arrangement. (This is indicated in the Supplementary Table). The reason for this is described in the Supplementary Material, Section “Arguments for choosing specific spatial-angular force distributions”, in the subsection *TVC cell pair*. Here is the argument:

“We have shown in Figure 1G that for two independent same cells moving together, the leading edge of the trailer cell is too wide, which suggests the protrusive force is too widely distributed, and the polarity of the cell pair is not well maintained. This implies that when two cells move together, their force distribution changes because of the cell-cell interactions. So, we decrease the protrusive force in the trailer cell and make it more concentrated along the central axis of the cell by multiplying the magnitude of the force by α. This is also consistent with the observation in vivo that in most cases, the two cells’ junction is concave if we look at it from the trailer cell.”

Which we highlighted for convenience. Lastly, the cell shapes and movements do not change significantly if the retraction forces in the leader or trailer are also graded with angle α, providing the derivative of the function of the angle is small enough.